

# Improved permafrost modeling in mountain environments by including air convection in a hydrological model

Gerardo Zegers[1], Masaki Hayashi[1], and Rodrigo Pérez-Illanes[2,3]

[1]Department of Earth, Energy and Environment, University of Calgary, Calgary, Canada
[2]Department of Civil and Environmental Engineering (DECA), Universitat Politècnica de Catalunya, Barcelona, Spain
[3]Hydrogeology Group (UPC-CSIC), Universitat Politècnica de Catalunya, Barcelona, Spain

**Correspondence:** Gerardo Zegers (gerardo.zegers@ucalgary.ca)

**Abstract.**

Permafrost occurrence in mountainous regions is influenced by complex topography and surficial geology, leading to high spatial heterogeneity. Coarse sediments create a unique thermal regime that allows permafrost to persist even under positive mean annual air temperatures due to natural convection lowering ground temperatures. Although this process has been rec-
ognized as a key factor in explaining the persistence of permafrost formation within coarse sediments, studies assessing the impact of natural convection on ground temperatures and permafrost are limited, partly due to the absence of hydrological models that take this process into account. This article expands on a well-established hydrological model to incorporate the effects of natural air convection on heat transfer. The modified model includes airflow through Darcy's equation and the Oberbeck-Boussinesq approximation to account for density-driven buoyancy effects, as well as a heat advection-conduction equation for
the air phase without assuming local thermal equilibrium between the air and the other phases. The model was tested on a talus slope in the Canadian Rockies, where conventional models failed to represent field-based evidence of permafrost. The results revealed that coarse-size sediments can lower ground temperatures by several degrees when natural convection is considered. Additionally, the study demonstrated that the local thermal equilibrium approach hinders the impact of natural convection. This enhanced model improves our understanding of permafrost dynamics in alpine landforms and enables a more accurate analysis
of permafrost extent and its influence on groundwater discharges.

## 1 Introduction

Mountains play a fundamental role in providing water resources for the environment and society, with nearly 40% of the global population relying on mountain runoff (Viviroli et al., 2020). These environments possess unique hydrological characteristics, sustaining streams throughout the year through diverse water storage processes—groundwater, lakes, snow, glaciers,
and permafrost—collectively supplying water across broad time scales (Jones et al., 2019). Within mountains, particular landforms like talus slopes, moraines, and rock glaciers stand out as key hydrological and ecological components of high-altitude basins. Recent field-based studies have shown that water released from these landforms contributes a relatively large fraction of the annual streamflow despite covering a relatively small area (Chen et al., 2018; Harrington et al., 2018; Hayashi, 2020; Somers et al., 2016; Winkler et al., 2016). Additionally, mountainous landforms exhibit unique thermal characteristics, where





the presence of coarse sediments creates a distinctive ground thermal regime, establishing them as cold-condition strongholds since their persistently cold discharges are critical for the ecological balance of mountain ecosystems (Brighenti et al., 2021; Harrington et al., 2017; Morard et al., 2008). Permafrost, defined as the subsurface material with temperatures below 0°C for at least two consecutive years (Riseborough et al., 2008), is likely to occur within these landforms, even under positive mean annual air temperature (MAAT) (Delaloye and Lambiel, 2005; Gorbunov et al., 2004; Haeberli et al., 2011; Wicky and Hauck,

2017). This context underscores the importance of understanding the thermal processes occurring in mountainous landforms as a necessary step to characterize the influence of permafrost in the hydrological balance of mountain environments.

    A relevant factor controlling permafrost occurrence in mountainous landforms is the presence of coarse, blocky sediments. Studies have shown that coarse sediments can sustain lower temperatures than adjacent fine-grained soils, primarily due to the combined influence of density-driven air convection and a lower thermal conductivity (Gorbunov et al., 2004; Gruber and

Hoelzle, 2008; Haeberli et al., 2011). During winter, large temperature gradients between the ground and the air can promote air convection inside high-permeability deposits, resulting in a strong coupling of the heat transfer processes between the atmosphere and the subsurface (e.g., Juliussen and Humlum, 2008; Pruessner et al., 2018). One example of this effect can be seen in landforms with steep slopes, such as talus slopes, where non-vertical air convection occurs during winter and summer, generating a thermal anomaly. While in general, air and ground surface temperatures decrease with elevation (Rolland, 2003),

internal air convection can override this relationship and lead to colder conditions in the lower part of a talus slope (e.g., Delaloye and Lambiel, 2005; Morard et al., 2008; Wagner et al., 2019; Wicky and Hauck, 2017). In addition, the coarse sediments' low thermal conductivity influences the seasonality of heat transfer, contributing to an overall decrease in ground temperatures (Gruber and Hoelzle, 2008). The heterogeneous nature of snowcover over these complex, non-smooth surfaces further influences heat transfer processes in coarse sediments, reducing the snow insulation effect (Haeberli et al., 2011).

Various numerical models have been developed to examine the complex interplay of processes influencing permafrost occurrence (Grenier et al., 2018). They have contributed to understanding the influence of groundwater flow in permafrost environments by coupling the flow and heat transport equations, including dynamic freeze-thaw processes. For example, SUTRA-ICE (McKenzie et al., 2007) and PFLOTRAN-ICE (Karra et al., 2014) are comprehensive subsurface three-dimensional (3D) models considering heat transport, water flow, and ice-related processes. However, they do not account for energy exchanges with

the atmosphere, requiring coupling with other models to simulate the surface energy balance and snowcover (e.g., Rush et al., 2021). In contrast, GEOtop, a well-known 3D model in mountain hydrology (Dall'Amico et al., 2018; Endrizzi et al., 2014; Fiddes et al., 2015; Rigon et al., 2006; Soltani et al., 2019), integrates surface and subsurface energy and water balances by solving the 3D Richards' equation to simulate subsurface flow and a 1D conduction equation for the energy balance. It is specifically designed to represent complex mountain terrains, considering the interaction between topography and radiation, a

feature not commonly found in other hydrological models.

    One major drawback of the models mentioned above is that none account for the necessary processes that lead to density-driven air convection, which restricts their usefulness in studying permafrost occurrence in coarse-grained landforms. While some studies have examined the influence of air convection on the thermal behavior of coarse sediments and permafrost occurrence (e.g., Arenson et al., 2006; Guodong et al., 2007; Kong et al., 2021; Lebeau and Konrad, 2016; Wicky and Hauck,





2020), these applications are often limited to two-dimensional problems and are not coupled with atmospheric processes. Furthermore, except for the work by Wicky and Hauck (2017, 2020), these studies are rarely discussed in the context of mountain environments. Models aiming to incorporate the effects of air convection commonly formulate the heat transport equation assuming local thermal equilibrium (LTE), meaning that the temperature of all the phases within the porous medium (sediment grains, water, ice and air) are assumed to be identical within the representative elementary volume. While the LTE

assumption might be suitable for modeling the advective water heat transport (due to its slower dynamics), it may not hold true for an accurate characterization of air convection, where the air can exhibit much faster dynamics (Nield and Bejan, 2017). Consequently, there is a need for a comprehensive cryo-hydrogeology model that integrates the physical processes influencing heat transport in coarse-grained landforms to improve the capability to predict and forecast permafrost occurrence.

Taking advantage of the versatility of the GEOtop model for hydrogeological studies in mountain environments, this article

extends GEOtop 3.0 to better represent the thermal behavior of coarse sediments in mountain environments by incorporating density-driven air convection into the model. The study specifically formulates the heat transport using a local thermal non-equilibrium (LTNE) approach. To the best of the authors' knowledge, this is the first instance of the LTNE approach being used in the context of permafrost modeling. The main objectives of this study are to (i) assess the validity of the commonly adopted local thermal equilibrium (LTE) approach for describing air convection within coarse sediments, (ii) enhance the understanding

of the impact of air convection on the energy balance of coarse sediments and to improve the capabilities of the GEOtop model to predict permafrost occurrence in mountainous landforms, and (iii) advance the understanding of permafrost's influence on the hydrological balance of talus slopes. To achieve these objectives, the study is based on the analysis of hypothetical test cases comparing the simulated ground temperatures of coarse sediments obtained with the existing GEOtop 3.0 and the new convection-enhanced GEOtop, with a particular focus on the implications of applying the LTNE approach. The new model is

further evaluated on a talus slope located in the Canadian Rockies, where field evidence of permafrost is available.

## 2  Model description

### 2.1  Natural convection

Natural convection, also known as density-driven convection, occurs when fluid flow is driven by density gradients, unlike forced convection, where an external force (e.g., pressure gradient) drives the flow. In coarse sediments, natural convection can

be generated when there is a large enough temperature gradient between the sediments and the external air. As a consequence, a density gradient develops between the lower (warmer) and upper (colder) boundaries, leading to the upward movement of warmer, less dense air and the downward movement of colder, denser air (Fig. 1). This pattern forms clockwise and counter-clockwise convection cells side by side, resulting in a bottom-up energy transfer by the convection cells (Narasimhan, 1999).

The strength of natural convection in a porous medium is quantified by the Rayleigh-Darcy number ($Ra$) (Bories and

Combarnous, 1973; Otero et al., 2004),

$$Ra = g \frac{\beta_a \rho_a c_a k}{\nu_a \lambda} \Delta T_a H, \tag{1}$$



where $g$ (m s$^{-2}$) is the gravitational acceleration, $\beta_a$ (K$^{-1}$) is the volumetric thermal expansion coefficient of the air, $\rho_a$ (kg m$^{-3}$), $c_a$ (J kg$^{-1}$ K$^{-1}$) and $\nu_a$ (m$^2$ s$^{-1}$) are, respectively, the density, specific heat capacity and kinematic viscosity of the air, $k$ (m$^2$) is the medium permeability, $\lambda$ (W m$^{-1}$ K$^{-1}$) the thermal conductivity of the medium and $\Delta T_a$ (K) is the difference

of air temperature between two plains separated by a distance $H$ (m). The onset of natural convection is defined by the critical Rayleigh number ($Ra_c$), which ranges from 27 to 40 depending on the upper surface characteristics (Lapwood, 1948). The lower limit can be considered to be representative of an open surface (e.g., coarse surface sediments without snow cover), whereas the upper limit can be taken to be representative of an upper surface impermeable to airflow (e.g., coarse surface sediments with thick snow cover).

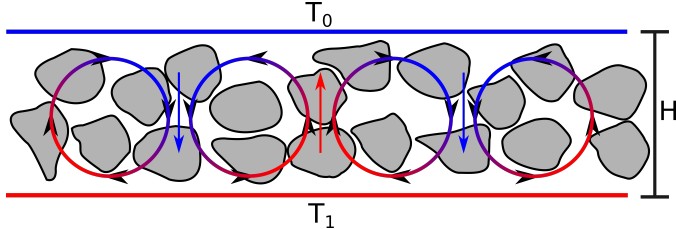

**Figure 1.** Density-driven air convection. The temperature at the bottom is higher than at the top ($T_1 > T_0$). Adapted from Guodong et al. (2007)

The Rayleigh number is particularly sensitive to the permeability of the porous medium, which can vary by several orders of magnitude for natural materials depending on the characteristic grain size (Nield and Bejan, 2017). Permeability is commonly estimated using the Kozeny-Carman equation,

$$k = A\,d_{10}^2\,\frac{n^3}{(1-n)^2}, \tag{2}$$

where $d_{10}$ (m) is a characteristic grain diameter such that 10% of the sediment particles are finer than $d_{10}$, $n$ is the porosity,

and $A = 0.0056/4.25$ is a constant proposed by Côté et al. (2011), that improves the estimation of permeabilities for coarse sediments.

Natural convection in coarse sediments has a strong seasonal variability resulting from air temperature fluctuations. In winter, temperatures inside the sediments are higher than the external air temperature. Air convection intensifies the energy exchange with the atmosphere, causing the sediments to lose energy to the air. In summer, the opposite happens; the temperature inside

the sediments is lower than the external air. Since the cold air is heavier than the exterior air, no air convection occurs, and energy exchange is controlled by conduction. The small contact areas between the sediments and the low thermal conductivity of the air result in the sediments acting as a thermal insulator, reducing the amount of energy gained by the ground from the air (Gruber and Hoelzle, 2008). Over the year, the imbalance of increased energy losses in winter and reduced energy gains in summer produces a net energy loss, lowering the average ground temperature (Guodong et al., 2007).





## 2.2 GEOtop

GEOtop is a versatile 3D hydrological model that effectively integrates ground energy and water budgets while accounting for atmospheric energy exchange and a multilayer snowpack (Bertoldi et al., 2006; Dall'Amico et al., 2011; Endrizzi et al., 2014; Rigon et al., 2006). This makes it suitable for modeling permafrost-relevant variables such as snow and ground temperatures. This work refers to GEOtop 3.0 as the standard GEOtop (S-GEOtop). The system of equations for the water flow and energy balance in S-GEOtop is as follows:

$$\frac{\partial \Phi_m}{\partial t} + \nabla \cdot \boldsymbol{J}_w + F_w = 0, \tag{3}$$

$$\frac{\partial U}{\partial t} + \frac{\partial G}{\partial z} = 0, \tag{4}$$

where $\Phi_m = \frac{M_w + M_i}{\rho_w V_c}$ is a dimensionless variable representing total water content expressed as liquid water, with $M_w$ and $M_i$ being respectively the mass of liquid water and ice, $\rho_w$ ($\mathrm{kg\,m^{-3}}$) is the density of liquid water, and $V_c$ ($\mathrm{m^3}$) is a control volume. $\boldsymbol{J}_w = -\frac{k_{rw}k}{\mu_w}(\nabla P - \rho_w \boldsymbol{g})$ ($\mathrm{m\,s^{-1}}$) is the water flux, $k_{rw}$ the relative permeability of water, $\mu_w$ ($\mathrm{Pa\,s}$) the dynamic viscosity of liquid water, $\nabla P$ ($\mathrm{Pa\,m^{-1}}$) is the pressure gradient, $\boldsymbol{g}$ ($\mathrm{m\,s^{-2}}$) is the gravity vector, and $F_w$ ($\mathrm{s^{-1}}$) is a sink term representing evapotranspiration. $U = C_m(T - T_{\mathrm{ref}}) + \rho_w L_f \theta_w$ ($\mathrm{J\,m^{-3}}$) is the internal energy, where $C_m$ ($\mathrm{J\,m^{-3}K^{-1}}$) is the medium volumetric heat capacity, $T$ (K) is the temperature within $V_c$, $T_{ref}$ is a reference temperature, $L_f$ ($\mathrm{J\,Kg^{-1}}$) is the latent heat of fusion, and $\theta_w$ is the volumetric liquid water content. $G = -\lambda_T \frac{\partial T}{\partial z}$ ($\mathrm{W\,m^{-2}}$) represents the vertical component of the conduction flux, where $\lambda_T$ ($\mathrm{W\,m^{-1}K^{-1}}$) is the effective thermal conductivity within $V_c$.

Prior research has utilized S-GEOtop in mountain regions to model snow cover (Dall'Amico et al., 2018), water and energy fluxes (Rigon et al., 2006; Soltani et al., 2019), and permafrost distribution (Fiddes et al., 2015; Wani et al., 2021). However, as the model does not consider airflow in porous media, it fails to accurately represent the thermal behavior of coarse sediments, leading to inaccurate representation of permafrost distribution in mountain landforms such as talus slopes, moraines, and rock glaciers.

## 2.3 Convection-enhanced GEOtop

### 2.3.1 Governing equations

To incorporate natural convection into the S-GEOtop model, a new set of transport equations is implemented to solve for the airflow within the sediments and the air heat transport, creating a new model version referred to as convection-enhanced GEOtop (CE-GEOtop). While the S-GEOtop formulation assumes that all phases are in thermal equilibrium within a control volume (Fig. 2A), CE-GEOtop uses the local thermal non-equilibrium (LTNE) approach to characterize the energy balance of the air phase (Leontini et al., 2015; Yang et al., 2012). Under this approach, the air temperature differs from the temperature of the other phases in the control volume (Fig. 2B). In this context, the group of all non-air phases will be referred to as the composite medium (CM). These phases are considered to be in local thermal equilibrium (LTE), as in the original implementation



of S-GEOtop, meaning that for each control volume, the new model will keep track of two different temperatures and the heat transfer processes between these two main phases. Although the LTNE approach requires additional parameters to determine the inter-phase heat transfer, it explicitly accounts for the decoupling between the temperature of the air phase and the CM, leading to a more realistic representation of convective heat transfer.

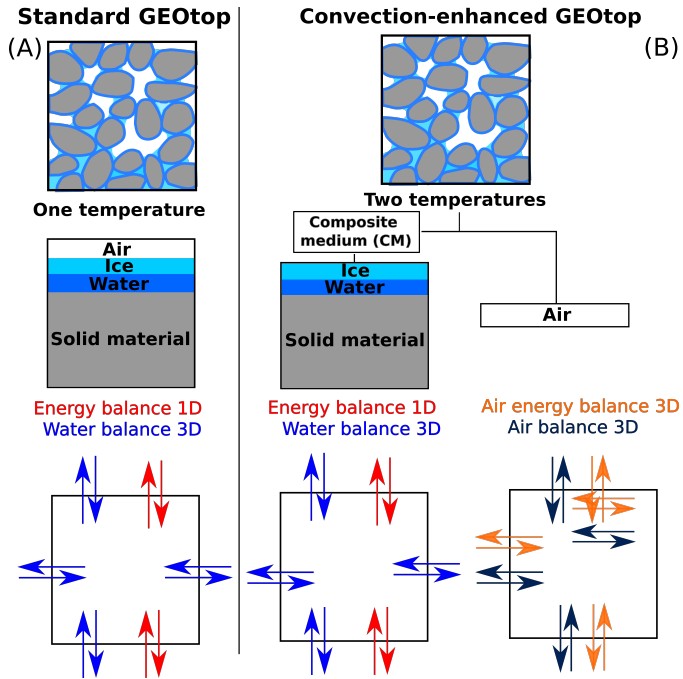

**Figure 2.** Panel (A) shows the conceptual model of the standard GEOtop model, and panel (B) shows the conceptual model of the convection-enhanced GEOtop.

Similarly to the water mass balance, the air mass balance is calculated using Richards' equation, with the important distinction that the air density now varies with temperature. This variation becomes relevant for natural convection, being the primary factor driving the airflow. The Oberbeck-Boussinesq approximation is used to solve the airflow equation, in which air is treated as an incompressible fluid (Eq. 5), and the density dependence is maintained only in Darcy's law (Barletta, 2009; Landman and Schotting, 2007). As detailed in Appendix A2, the airflow equations are expressed as:

$$\rho_a \nabla \cdot \boldsymbol{V}_a = 0, \tag{5}$$

$$\boldsymbol{V}_a = \frac{k_{ra}k}{\mu_a}(\nabla P - \rho_a \mathbf{g}), \tag{6}$$

where $\boldsymbol{V}_a$ $(\mathrm{m\,s^{-1}})$ is the air velocity, $k_{ra}$ is the relative permeability of the air, and $\mu_a$ $(\mathrm{Pa\,s})$ is the dynamic viscosity of air. The air density is calculated as $\rho_a(T_a) = \rho_a(T_{ref})\left[1 - \beta \cdot (T_a - T_{ref})\right]$ (Nield and Bejan, 2017). $k_{ra}$ is calculated by the van



Genuchten approach as proposed by Mahabadi et al. (2016), which uses the same parameters for both the water and air relative permeabilities:

$$k_{ra} = (1 - \bar{S}_a)^{0.5}\left[1 - \bar{S}_a^{1/m}\right]^{2m}, \quad \bar{S}_a = \frac{S_w - S_{rw}}{1 - S_{rw}}, \tag{7}$$

where $S_{rw}$ is the residual water saturation, $m$ is a van Genuchten parameter, and $S_w = \frac{\theta_w}{n}$ is the relative liquid water content. For solving equation 5, the lateral and bottom domain boundaries are assumed to be impermeable, whereas the top boundary

condition is snow-dependent; in snow-free pixels, an open boundary condition is applied, allowing air exchange between the soil and the atmosphere, while a no-flux boundary is applied to snow-covered pixels. Some authors have reported air infiltration in the soil even with snow cover up to 0.5-1 m (Delaloye and Lambiel, 2005; Morard et al., 2008; Scherler et al., 2013); therefore, a threshold parameter, $H_{ai}$ (m), is included to account for potential air infiltration in snow-covered pixels. For snow depths below $H_{ai}$, the airflow remains unimpeded by snow cover (open boundary condition).

The energy transport equation in the air phase combines advection and conduction processes, but it does not account for the thermal radiation of coarse sediments:

$$\frac{\partial U_a}{\partial t} + \nabla \cdot (\boldsymbol{G_a} + \boldsymbol{J_a}) - h(T_{cm} - T_a) = 0, \tag{8}$$

where $U_a = C_a(T_a - T_{ref})$ (Jm$^{-3}$) is the air internal energy, $T_a$ (K) and $T_{cm}$ (K) are the temperatures of the air and the composite medium respectively, $C_a$ (J m$^{-3}$K$^{-1}$) is the air volumetric heat capacity, and $\theta_a$ is the air content. The heat conduction

flux is denoted by $\boldsymbol{G_a} = -\lambda_a \nabla T_a$ (Wm$^{-2}$), where $\lambda_a$ (Wm$^{-1}$K$^{-1}$) is the air thermal conductivity, while the heat advected by the airflow is given by $\boldsymbol{J_a} = \rho_a C_a(T_a - T_{ref})\boldsymbol{V_a}$ (Wm$^{-2}$). Lastly, $h$ (Wm$^{-3}$K$^{-1}$) is a heat transfer coefficient describing the heat exchange between the air and the composite medium. For a proper representation of the coupling between the two phases, the heat exchange term included in equation 8 should be included with the opposite sign in the energy balance for the composite medium (Eq. 4). Following the approach proposed by Leontini et al. (2015), the heat transfer coefficient $h$ is

calculated as,

$$h = \frac{A_p h_{cm-a}(1 - \theta_a)}{V_p}, \tag{9}$$

where $A_p$ (m$^2$) and $V_p$ (m$^3$) are respectively the surface area and volume of a spherical particle with diameter $d_p$, and $h_{cm-a}$ (Wm$^{-2}$K$^{-1}$) is the heat transfer coefficient between the CM and the air interface calculated as,

$$h_{cm-a} = \frac{Nu_{sa}\lambda_a}{d_p}. \tag{10}$$

In the previous expression, $Nu_{sa}$ is the solid-to-air Nusselt number, which is expressed as

$$Nu_{sa} = a + b \cdot Pr_a^{1/3} \cdot Re_p^c. \tag{11}$$

where, $Re_p = \frac{|\boldsymbol{V_a}| d_p}{\nu_a}$ is the particle Reynolds number, and $Pr_a$ is the air Prandtl number. At atmospheric pressure, $Pr_a$ is fairly constant for temperatures between -10 to 25 °C, so a constant value of $Pr_a = 0.711$ is used, corresponding to its value





for 0 °C. In Eq. 11, the coefficients $a$, $b$, and $c$ are the heat transfer model constants; for packed spheres, were estimated to be $a = 2$, $b = 0.5$, and $c = 1/3$ (Whitaker, 1977).

### 2.3.2 Thermal conductivity

Thermal conductivity plays a fundamental role in the sediment's conductive heat transport and, thus, in the exchange of heat with the atmosphere. Particularly, the low thermal conductivity of coarse sediments leads to an overall decrease in ground temperatures (Gruber and Hoelzle, 2008). Currently, S-GEOtop calculates the sediment's effective thermal conductivity $\lambda_T$, using the approach proposed by Cosenza et al. (2003):

$$\lambda_T = [(1-n)\sqrt{\lambda_s} + \theta_w\sqrt{\lambda_w} + \theta_i\sqrt{\lambda_i} + \theta_a\sqrt{\lambda_a}]^2 \tag{12}$$

where $\theta_i$ is the volumetric fraction of ice, $\lambda_s$ the thermal conductivity of sediment particles, $\lambda_w$ the thermal conductivity of water, and $\lambda_i$ the thermal conductivity of ice. This approach generally leads to a reasonable estimation of the sediments' thermal conductivity, although it overestimates the conductivity in dry conditions. Using Eq. 12, the minimum thermal conductivity is $\lambda_T = [(1-n)\sqrt{\lambda_s} + \theta_a\sqrt{\lambda_a}]^2 \sim 0.9\ \mathrm{Wm^{-1}K^{-1}}$ for a case with negligible water content, porosity of 40% and solid thermal conductivity of 2.2 $\mathrm{Wm^{-1}K^{-1}}$. However, previous authors have shown that for liquid water contents lower than 5-10%, $\lambda_T \sim 0.3 - 0.5\ \mathrm{W\,m^{-1}\,K^{-1}}$ (Bristow, 2018; Côté and Konrad, 2009), that is about the half of the value estimated by using Eq. 12. These low thermal conductivities during dry conditions occur since the rock-to-rock contacts are minimal due to the rock roughness (Cheng and Wong, 2022). Those small open spaces on the rock-to-rock contact are filled with air for dry conditions, hindering thermal conduction (Bristow, 2018).

In CE-GEOtop, a new approach to calculating the thermal conductivity was implemented. Particularly, the estimation of the thermal conductivity for low water content conditions is improved by using the Farouki-de Vries model (Farouki, 1981; Tian et al., 2016):

$$\lambda_T = \frac{\theta_w \cdot \lambda_w + w_i \cdot \theta_i \cdot \lambda_i + w_a \cdot \theta_a \cdot \lambda_a + w_s \cdot (1-n) \cdot \lambda_s}{\theta_w + w_i \cdot \theta_i + w_a \cdot f_a + w_s \cdot (1-n)} \tag{13}$$

where $w_n$ are the weighting factors of air (a), ice (i), and solid particles (s) that can be calculated by:

$$w_n = \frac{2 \cdot g_n}{3}\left[1 + \left(\frac{\lambda_n}{\lambda_w} - 1\right)\right]^{-1} + \frac{1}{3}\left[1 + \left(\frac{\lambda_n}{\lambda_w} - 1\right)(1 - 2g_n)\right]^{-1} \tag{14}$$

where $g_n$ are the shape factors for each phase. A uniform shape factor $g_n = 0.125$ is used for both sediment particles and ice and the shape factor of air ($g_a$) depends on the water content,

$$g_a = \begin{cases} 0.333 - (0.333 - 0.035)\theta_a/n, & \text{if } \theta_w \geq 0.09 \\ 0.013 + 0.944\theta_w. & \text{if } \theta_w < 0.09 \end{cases} \tag{15}$$

The Farouki-de Vries model sets liquid water as the continuum medium and sediment minerals as uniform particles to estimate $\lambda_T$ of unfrozen and frozen soil. Under the same dry conditions mentioned earlier, the Farouki-de Vries will predict a



thermal conductivity of $\lambda_T = 0.32$ Wm$^{-1}$K$^{-1}$, which is consistent with the reported sediment's thermal conductivity for dry conditions ($\lambda_T \sim 0.3 - 0.5$ W m$^{-1}$ K$^{-1}$; Bristow, 2018; Côté and Konrad, 2009).

### 2.3.3 Numerical Solver

The four main transport equations are solved sequentially in a time-lagged manner (Panday and Huyakorn, 2004): (i) air mass balance (Eq. 5), (ii) air energy balance (Eq. 8), (iii) composite medium energy balance (Eq. 4) and, (iv) water mass balance (Eq. 3). Each equation is solved by using the implicit finite-volume scheme originally implemented in S-GEOtop. The solver is based on a Newton-Raphson iterative scheme (more details on Appendix A1).

Given the nonlinear nature of the system of equations, maintaining control over the numerical convergence of the simulation

becomes crucial. Arenson et al. (2006), in their study involving air convection simulations, underscored the challenge of establishing an appropriate relationship between the medium permeability and the numerical resolution parameters (time step and mesh spacing). They emphasized that numerical instabilities may arise rapidly when these factors are not adequately balanced.

In the CE-GEOtop, a mechanism has been implemented to monitor the stability of the numerical solution by using the

maximum grid Courant number for the air velocities ($Cu$):

$$Cu = \max\left(\left\{\Delta t \frac{\boldsymbol{V}_a}{\Delta d}\big|_i, i =, \cdots, N\right\}\right) \tag{16}$$

where $\boldsymbol{V}_a|_i$, and $\Delta d|_i$ are respectively the air velocity and spacing at the $ith$ face of the grid, and $N$ is the number of faces in the grid. At each iteration, the model examines the $Cu$ and reduces the $\Delta t$ if $Cu$ surpasses the user-defined maximum value, $Cu_{max}$. Despite the utilization of an implicit numerical scheme, it is generally advised to maintain $Cu_{max} < 10$ to ensure both

stability and accuracy of the numerical solution (Rai and Chakravarthy, 1986).

### 2.3.4 Shadows

Radiation plays an important role in the ground-surface energy balance. In complex terrains, careful considerations are necessary for an accurate estimation of the total radiation that the surface receives. S-GEOtop accounts for both cast shadows from surrounding topography and the view factor's impact on radiation (Endrizzi et al., 2014). However, during the development

of the new code, it was detected that the method for calculating cast shadows could be improved to provide a more accurate estimation of the shadows in rugged terrains. The original approach in S-GEOtop uses the model digital elevation mode (DEM) to calculate cast shadows, but it often fails to account for shadows cast from peaks outside the model's domain. In CE-GEOtop, this limitation is addressed by allowing for the input of a second, larger DEM specifically for calculating cast shadows. This enhancement improves the model's capacity to represent shadow conditions without expanding the model's dimensions.



## 3   Case study - Babylon talus slope

To evaluate CE-GEOtop's performance in a realistic scenario, the model is applied to the Babylon talus slope in the Lake O'Hara watershed, Yoho National Park, Canada, where field evidence suggests permafrost presence (Fig. 3). The Babylon basin is approximately 0.28 km$^2$ in surface area and is drained by Babylon Creek, which exits the northeast corner of the basin. Meteorological data from a nearby weather station show that the mean monthly air temperature in the area ranges from -9.7 °C in January to 10.4 °C in July, with annual precipitation of 1000-1200 mm, predominantly in the form of snow (He and Hayashi, 2019). Two small springs at the base of the Babylon talus flow toward Babylon Creek along the bedrock plane. Discharge and temperature were measured weekly at Babylon Creek from July to September 2008, with continuous stage recording every 10 minutes using pressure transducers in a stilling well (AquaTroll 200, In-Situ Inc.; Muir et al., 2011). Discharge was estimated using a stage-discharge rating curve developed from manual water stage measurements.

The bedrock underlying the talus is from the Lower Cambrian Gog Group, mainly consisting of quartzite, while Middle Cambrian carbonates cap the peaks surrounding the watershed (Price et al., 1980). Geophysical surveys and hydrograph analysis indicate that the talus has an openwork matrix with large void spaces without the presence of fine material (Muir et al., 2011).

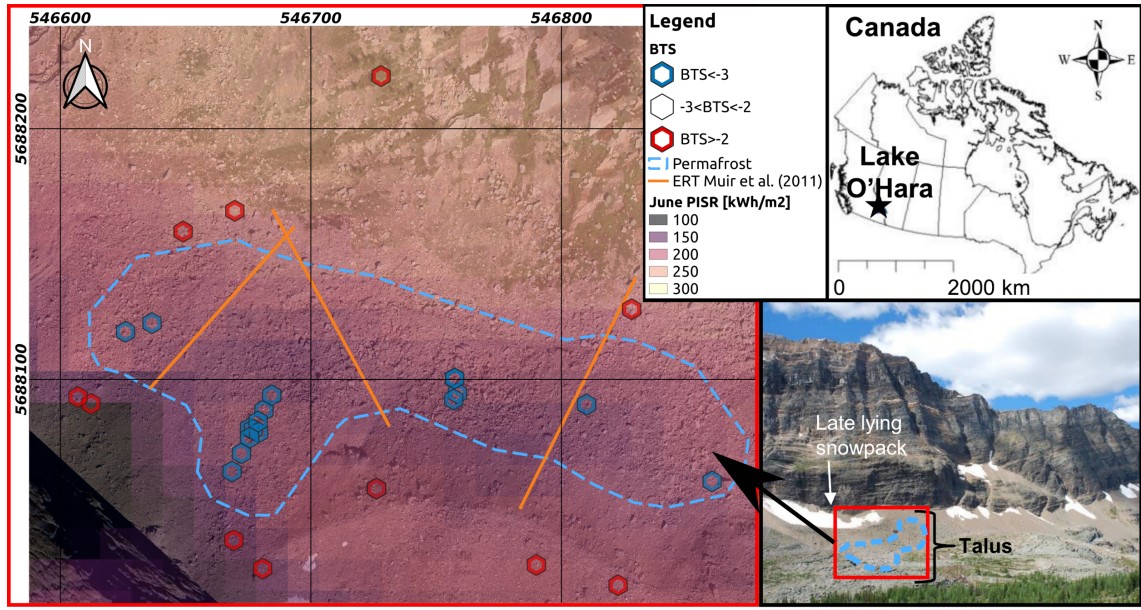

**Figure 3.** Map of Babylon talus slope showing the locations of BTS sensors (hexagons), electrical resistivity tomography profiles (orange lines), and estimated permafrost extent (blue line). June potential incoming solar radiation (PISR) was calculated using SAGA GIS (version 2.3.1; http://saga-gis.org) and the satellite digital elevation model ALOS PALSAR (12x12 m: JAXA/METI ALOS PALSAR L1.0 2007).

In specific sections of the Babylon talus slope, high electrical resistivity values exceeding 50 kΩm were observed (Muir et al., 2011), indicating the presence of permafrost (Hauck and Kneisel, 2008). To verify this, surface temperature sensors



(Ibuttons DL1925 and DS1921Z-F5) were deployed during the winter seasons of 2016, 2021, and 2022. By correlating high resistivity zones with low basal temperatures of the snowpack (BTS), permafrost was identified in the middle-bottom section of the Babylon talus slope (Fig. 3). The presence of permafrost in the middle-bottom section of the talus and within coarser sediments suggests that natural convection influences permafrost occurrence at the Babylon talus slope, making it an interesting

case study for evaluating CE-GEOtop under actual field conditions.

## 4    Results and discussions

Thoroughly evaluating the CE-GEOtop model presents a challenge due to the absence of an analytical solution encompassing all the processes the model accounts for. Nonetheless, the standard GEOtop model has been validated by previous studies (Dall'Amico et al., 2011; Endrizzi et al., 2014); hence, this study will focus on a comparative analysis of the new air convection

module in CE-GEOtop with respect to equivalent simulations performed with S-GEOtop. This section begins by examining the effects of local thermal non-equilibrium (LTNE) using a synthetic case. Following this, the performance of the air convection module is assessed on the Babylon talus slope.

### 4.1    Effect of Rayleigh number and Local Thermal Non-Equilibrium (LTNE).

Simulations using CE-GEOtop were conducted to evaluate the differences between a case considering the classical LTE as-

sumption against a case using the LTNE approach. These simulations evaluated the impact of considering different heat transport processes in the sediment's temperature without considering surface energy fluxes. The LTE simulations were achieved by artificially increasing the heat transfer term ($h$; Eq. 9), thus enforcing the same temperature between the air and the composite medium. Additionally, S-GEOtop simulations were performed to establish a baseline for comparison with CE-GEOtop. Since S-GEOtop exclusively considers conduction as the heat transport mechanism, any agreement between the results of

CE-GEOtop and S-GEOtop implies the absence of natural convection.

The numerical experiment involved a synthetic geometry representing a square flat domain with dimensions of 33 m by 33 m and a 3 m layer of coarse sediments with a permeability of $1.0 \times 10^{-6}$ m$^2$ ($d_{10} \sim 70$ mm). The domain was discretized into 12 vertical layers of 0.25 m each and a grid size of 1.5 m, resulting in 22 grid cells in both the longitudinal and transversal directions.

To limit the interaction with the atmosphere, the surface energy balance was removed, which included eliminating shortwave and longwave radiation (by setting surface albedo to 1 and surface emissivity to 0), latent heat (by setting sediment water content to $\theta_w = 0$ and using a low air relative humidity), and sensible heat (by setting a low wind velocity). The sediments were initialized with a temperature of 15 °C, and a no-flux lower boundary condition was applied for the heat transfer. On the top boundary, a fixed temperature ($T_{top}$) was applied to the sediments and air throughout the simulation. The simulations were

carried out for various values of $T_{top}$, resulting in a range of Rayleigh numbers from 25 to 265. With a lower impermeable boundary and an upper open surface, $Ra_c$ was estimated to be 27 (Lapwood, 1948). The simulations were conducted with





constant boundary conditions over a period of 30 days. The results were analyzed in terms of the layer-averaged normalized temperature $T_t^*(z)$:

$$T_t^*(z) = \frac{\overline{T_t(z)} - T_{top}}{T_{ini} - T_{top}} \tag{17}$$

where $\overline{T_t(z)}$ (K) is the temperature average over the layer at time $t$ (s) and $T_{ini}$ (K) is the initial temperature.

All simulations exhibit identical temperature profiles for the lowest Rayleigh ($Ra = 25$), indicating the absence of natural convection (Fig. 4). However, with a slightly higher Rayleigh number ($Ra = 65$), natural convection influences the sediment temperature profiles after approximately 15-20 days in the LTNE simulation. This suggests that although the Rayleigh number exceeds its critical value, natural convection is not strong enough to affect ground temperatures over short periods. Analysis of
the results of the higher $Ra$ values reveals that natural convection becomes increasingly significant for $Ra > 100$ as evidenced by the temperature profiles while considering the LTNE approach.

At monthly time scales, the LTE simulations are indistinguishable from the pure conduction simulations in all cases except for $Ra = 265$. In this last case with the highest Rayleigh number, LTE simulations show lower temperatures than the pure conduction case after 20 days and eventually resemble the LTNE simulations after 30 days, indicating that the system reaches local
thermal equilibrium. However, significant disparities are observed for shorter times or lower Rayleigh numbers, highlighting the system's non-thermal equilibrium state. This is evident in the air temperature profiles for the LTNE case (dashed cyan lines), where early air temperature indicates lower temperatures than the sediment. As a result, the thermal non-equilibrium state leads to a faster decrease in the temperature of the sediments. Therefore, adopting the LTE approach affects the short-term response of the sediments' temperature and hinders natural convection effects.

## 4.2   Thermal anomalies at talus slopes

Topoclimatic factors, such as air temperature and potential incoming solar radiation (PISR), are typically used to delineate permafrost zones. For the Babylon talus slope case study, these variables would suggest that the upper part of the talus is the most favorable zone for the occurrence of permafrost due to the lower air temperatures and PISR. However, the observed permafrost zone is in the middle-bottom part of the talus (Fig. 3), similar to observations in previous studies elsewhere (e.g.,
Delaloye and Lambiel, 2005; Morard et al., 2008; Phillips et al., 2009; Popescu et al., 2017). This thermal anomaly could be caused by air convection in two characteristic regimes: the winter regime, in which cold external air induces the influx of air into the bottom of the slope, and warm air outflow at the top (Fig. 5A); and the summer regime, in which higher external air temperature causes the influx of warm air at the top and the outflow of cold air at the bottom (Fig. 5B)

These two regimes (summer and winter) will be simulated to assess whether the CE-GEOtop model can realistically rep-
resent the thermal behavior of talus slopes. CE-GEOtop (with LTNE) and S-GEOtop models were configured using the dimensions of the Babylon talus slope with a constant slope angle of 21 degrees derived from a DEM. The 3D model domain is confined within the boundaries of the geophysical survey (Muir et al., 2011), with the talus slope's longitudinal axis measuring 94.5 m and the transversal axis 38.5 m. Figure 5C shows a cross-section of the model along the longitudinal centerline of the



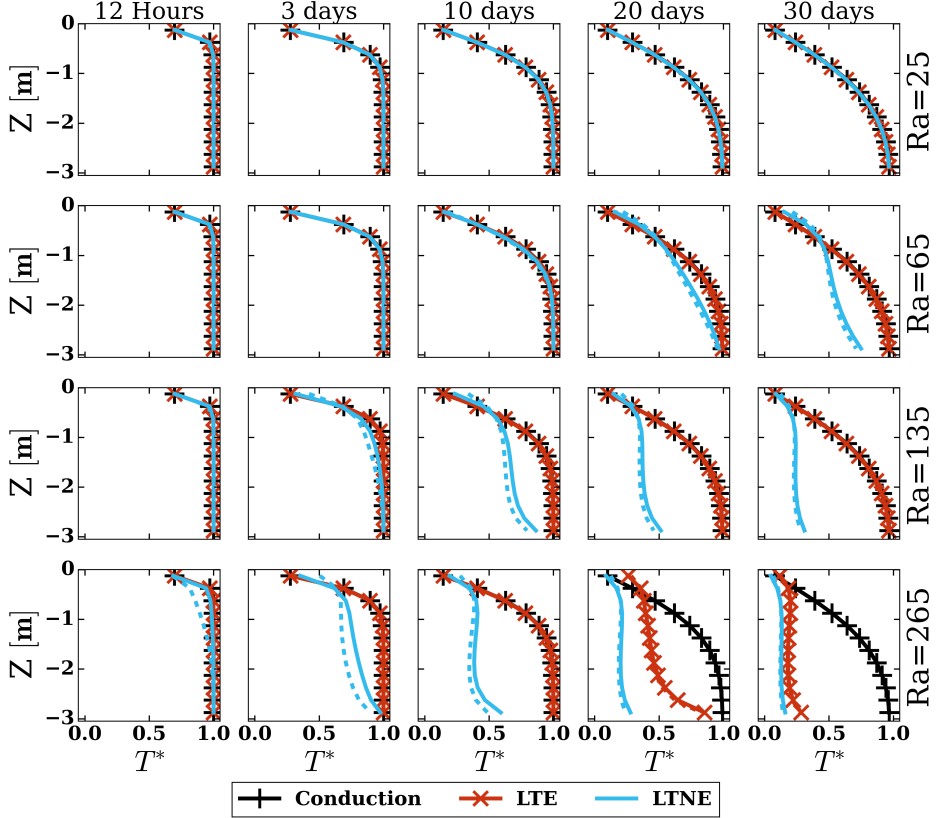

**Figure 4.** Layer-average temperature profiles for different $Ra$ values plotted against normalized temperature ($T^*$) for S-GEOtop (conduction only), CE-GEOtop (LTE), and CE-GEOtop (LTNE). The solid lines represent the sediment temperature, and the dashed lines represent the air temperature.

slope. The grid size is set at 3.5 m, resulting in 27 grid cells in the longitudinal direction and 12 in the transversal direction.

The bedrock depth, based on Muir et al. (2011), reaches a maximum of 16 m at the middle section of the talus slope, gradually decreasing to 0 m at the top and bottom regions. Consequently, the initial 16 m of the sediment domain is discretized into 32 layers, each one with a thickness of 0.5 m, while the subsequent 12 layers representing the bedrock have varying thicknesses ranging from 0.5 m to 2 m, in total reaching to a depth of 33.25 m. A constant geothermal heat flux of 0.03 W m$^{-2}$ was set as the model's lower boundary condition (Scherler et al., 2013).

For each regime, three cases were run; the first, CG-HK, used the CE-GEOtop model with coarse sediments and a high permeability of $k = 1.5 \times 10^{-6}$ m$^2$ representing low-end values for cobble-size materials based on laboratory measurements (Côté et al., 2011). The second is SG-HK, which utilized the standard GEOtop model with coarse sediments ($k = 1.5 \times 10^{-6}$ m$^2$). The last one is CG-LK, employing CE-GEOtop with sand-size sediments with low permeability ($k = 1 \times 10^{-10}$ m$^2$)



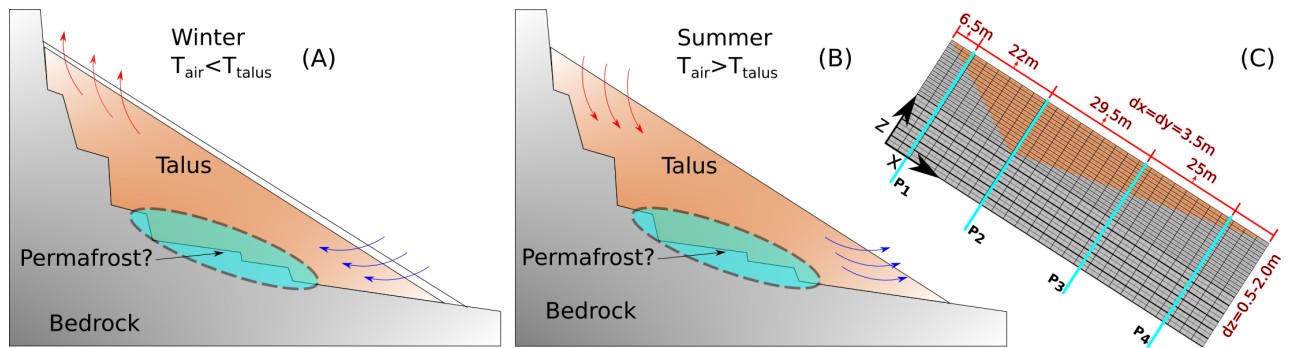

**Figure 5.** Conceptual model of the talus slope: (A) winter regime and (B) summer regime (modified from Morard et al. (2008)); and (C) a cross-section of the 3D model along the longitudinal line running in the center of the slope (y = 19.25 m), where vertical lines show the positions of temperature profiles ($P_1$, $P_2$, $P_3$, and $P_4$).

instead of coarse sediments. For all cases, bedrock is assumed to be almost impermeable with a $k = 2 \times 10^{-15}$ m$^2$ (Hayashi,

335    2020).

Simulations of summer and winter regimes were conducted with constant boundary conditions over a duration of 10 days. Similar to the cases analyzed in the previous section, all surface energy balance terms were neglected.

### 4.2.1    Winter regime

The talus and bedrock's initial temperature were set to $T = 10$ °C, whereas the external air temperature was set to 2.5 °C.

Figure 6 compares the simulated temperatures at four different profiles located along the center line of the model domain: $P_1$, $P_2$, $P_3$, and $P_4$ (see Fig. 5C for location).

In the SG-HK case, all temperature profiles show similar behavior due to heat transfer restricted to one-dimensional vertical heat conduction. After 10 days, a slight surface cooling of about 1.5 °C is observed. Similarly, there is a small temperature drop in the CG-LK case, and temperatures are very similar in magnitude to those obtained in the SG-HK case, except slightly

higher at the surface. The differences between CG-LK and SG-HK at the surface can be attributed to the different thermal conductivity calculations, with SG-HK overestimating the thermal conductivity of dry sediments (as outlined in section 2.3.2). Conversely, the CG-HK case, which considers coarse sediments, displays distinctive differences among the four profiles. The upper section ($P_1$) exhibits higher temperatures, while a more pronounced cooling is evident in the middle ($P_2$ and $P_3$) and bottom ($P_4$) sections. The middle section cools by 7 °C and the bottom section by 6.5 °C, with lower temperatures penetrating

to depths of 4-6 m.

In the CG-HK case, the airflow streamlines indicate that the temperature distributions are influenced by natural convection (Fig. 7). Initially, cold air infiltrates mainly at the bottom of the talus slope, pushing out warmer air at the top, leading to the most substantial temperature drop in the lower section (Fig. 7A). As the simulation progresses, the primary infiltration area shifts to the middle, forming a V-shaped pattern in the streamlines. The perpendicular nature of the air infiltration in this section





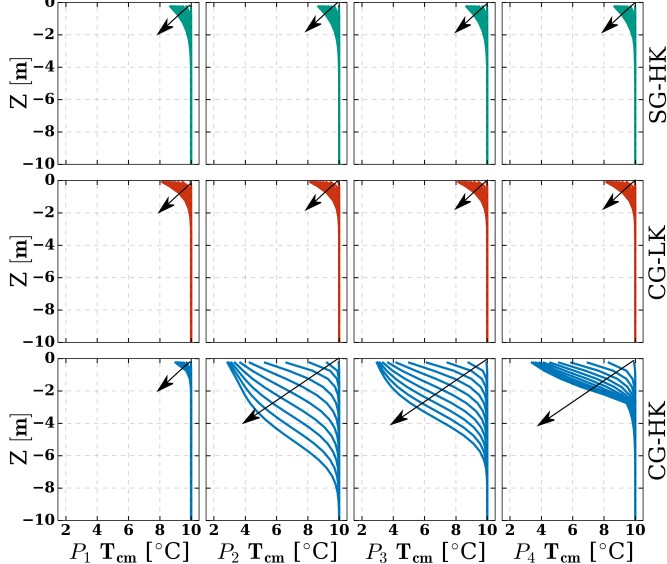

**Figure 6.** Daily sediment temperature profiles for the winter regime up to 10 days. Arrows indicate time progression from beginning (right) to end (left). Green lines represent the SG-HK case (S-GEOtop with high permeability), red lines represent the CG-LK case (CE-GEOtop with low permeability), and blue lines represent the CG-HK case (CE-GEOtop with high permeability). Profiles at $P_1$, $P_2$, $P_3$, and $P_4$ correspond to the upper, middle, and bottom sections of the talus slope (Fig. 5C).

allows for the cold front to penetrate deeper into the slope. This flow pattern correlates with the talus slope's geometry, where the middle part, having the highest Rayleigh number (Eq. 1), exhibits the greatest potential for natural convection and airflow. These results align with previous studies on talus slopes (Morard et al., 2008; Popescu et al., 2017), indicating that cold air enters the middle-bottom section and warm air rises to the top. This explains the temperature distribution in Figure 6, where cold air infiltration decreases temperatures in the middle-bottom section while warmer air maintains a stable temperature in the top section.

#### 4.2.2 Summer regime

To simulate the summer regime, the initial temperature of the talus and bedrock was set to 10 °C, while the external air temperature was set to 17.5 °C. All temperature profiles show similar patterns for the SG-HK case (see Fig. 8), with a surface temperature increase of approximately 1.5 °C after 10 days. The CG-LK case once again demonstrates a lack of natural convection in lower permeability sediments, leading to temperature profiles similar to those of the SG-HK case. Conversely, the CG-HK scenario predicts a slight temperature increase at the bottom section surface and a stronger increase at the top section surface (5 °C). Interestingly, the higher temperatures in the middle-top section of the talus for this case suggest that convection creates less favorable conditions for permafrost occurrence.



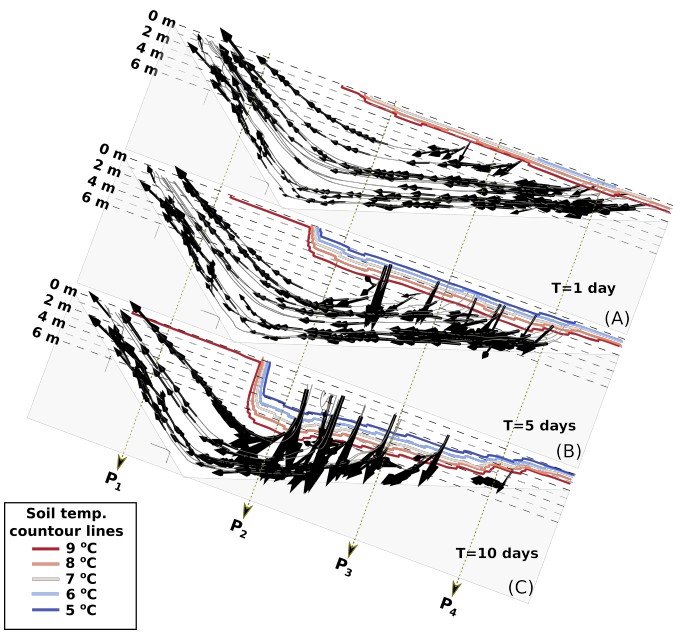

**Figure 7.** Airflow streamlines in the talus slope for the case CG-HK during the winter phase after 1, 5, and 10 days of simulation. The grey lines show the air streamlines, while the arrows represent the direction and relative magnitude of the airflow velocity. The colored lines represent the sediment temperature contour lines.

The airflow streamlines for the summer CG-HK case reveal that the temperature distributions are explained by a reversal of the airflow that goes from the top to the bottom of the talus (Fig. 9). This reversal leads to an increase in temperature in the upper section due to warm air infiltration, while the lower zone maintains a temperature of around 10-11 °C throughout the simulation as air flows from within the talus slope. This flow pattern supports the "cold reservoir" observations, where a cooler airflow persists in summer, keeping some middle-lower zones at low temperatures (Morard et al., 2008; Popescu et al., 2017).

### 4.3 Permafrost predictions

This section elaborates on the CE-GEOtop model's capability to predict permafrost distribution within the Babylon talus slope. The model is forced with half-hourly meteorological data from the nearby ($\sim$0.5 km) Opabin weather station (He and Hayashi, 2019), including precipitation, wind speed and direction, relative humidity, air temperature, incoming longwave radiation, and cloud fraction (estimated from incoming shortwave radiation data (Long et al., 2006)). Spatial distribution of precipitation, relative humidity, and air temperature was achieved using lapse rates calculated by Hood (2013) from measurements along an
elevation transect in the local area.

The model's dimensions, including spatial discretization and the bedrock depth, are consistent with the setup described in the preceding section (Fig. 5C). However, this analysis incorporates distinct sediment permeability and thermal properties. Based on the findings of Kurylyk and Hayashi (2017), a low conductivity layer ($k = 3.75 \times 10^{-11}$ m$^2$) was incorporated in the bottom



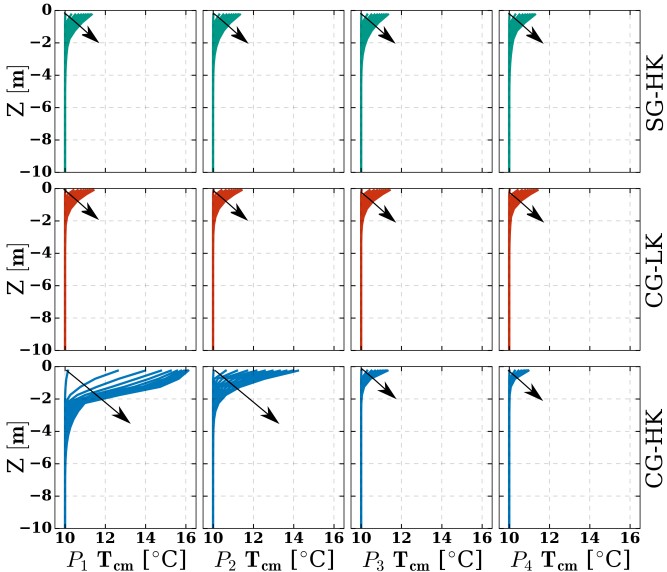

**Figure 8.** Daily sediment temperature profiles for the summer regime up to 10 days. Arrows indicate time progression from beginning (left) to end (right). Green lines represent the SG-HK case (S-GEOtop with high permeability), red lines represent the CG-LK case (CE-GEOtop with low permeability), and blue lines represent the CG-HK case (CE-GEOtop with high permeability). Profiles at $P_1$, $P_2$, $P_3$, and $P_4$ correspond to the upper, middle, and bottom sections of the talus slope (Fig. 5C).

of the talus sediments. Given that the talus is predominantly composed of cobble to boulder-sized rocks and lacks fine sediments

(Muir et al., 2011), a uniform permeability of $1.5 \times 10^{-6}$ m$^2$ ($d_{10} = 80\,mm$) was chosen to represent the entire unit, which is likely conservative considering the larger sediments observed in the field. The sediment lithology is assumed to mirror the overlying rockwall, with approximately 65% quartzite and 35% limestone (Price et al., 1980). Quartzite possesses a thermal conductivity of $\lambda_T = 5.7$ Wm$^{-1}$K$^{-1}$ (Jones, 2003), while limestone has a thermal conductivity of $\lambda_T = 2.9$ Wm$^{-1}$K$^{-1}$ (Thomas et al., 1973). Additionally, the volumetric heat capacity of limestone ($2.1 \times 10^6$ Jm$^{-3}$K$^{-1}$; Shen et al. (2018)) is

slightly higher than that of quartzite ($1.9 \times 10^6$ Jm$^{-3}$K$^{-1}$; Jones (2003)). Consequently, the thermal properties of the talus sediments were calculated proportionally to the quantities of quartzite and limestone.

    The presence of snow has a strong effect on sediments' temperature. To verify the accuracy of CE- GEOtop's simulated snow water equivalent (SWE), it was compared to the snowcover evolution estimated by Hood and Hayashi (2015) in the Opabin watershed, which includes the Babylon talus slope. Hood and Hayashi (2015) validated the Utah Energy Balance (UEB)

model using extensive field campaigns. The Babylon talus slope falls within the regions covered by oblique-angle terrestrial photography used for UEB model validation, so it is aimed to match the snow cover period rather than the exact SWE values.

    The surface aerodynamic roughness was set to 20 mm, and the rain threshold temperature to 1 °C in CE-GEOtop (Hood and Hayashi, 2015). While Hood and Hayashi (2015) simulations began in mid-April with measured snow water equivalent (SWE) as the starting condition, CE-GEOtop simulations, driven solely by meteorological data, spanned two years to mitigate the





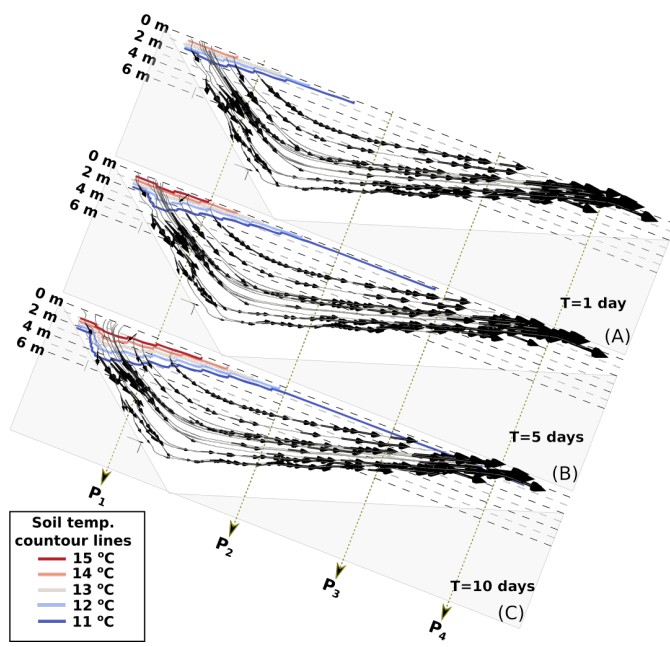

**Figure 9.** Airflow streamlines in the talus slope for the case CG-HK during the summer phase after 1, 5, and 10 days of simulation. The grey lines show the air streamlines, while the arrows represent the direction and magnitude of the airflow velocity.

uncertainty in initial conditions. Figure 10B compares SWE from CE-GEOtop and UEB, showing similar snow cover patterns, particularly in terms of the timing of snow-free ground, despite different methodologies. This highlights CE-GEOtop's ability to simulate SWE dynamics effectively in the Babylon talus slope, reinforcing its value in permafrost research.

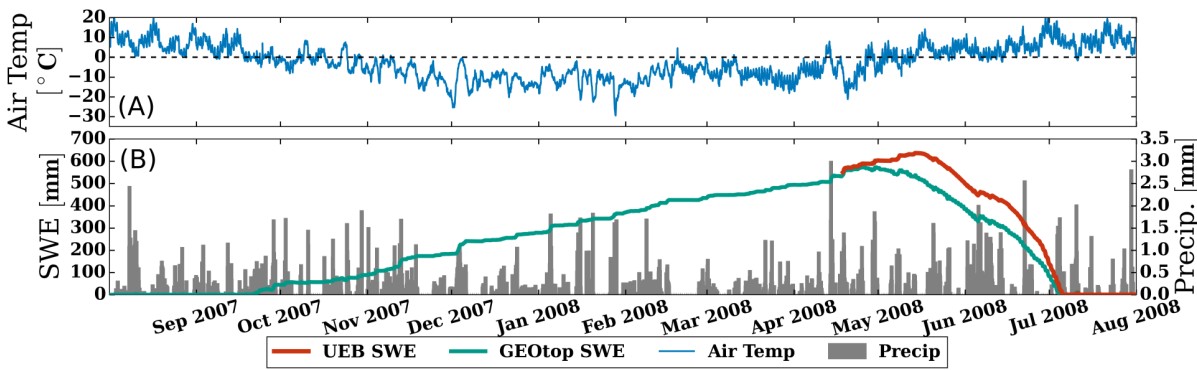

**Figure 10.** Panel (A) shows the observed air temperature. The left axis of B) shows the simulated SWE, where the green line represents the CE-GEOtop simulated SWE, while the red line represents the UEB simulated SWE. Panel B) on the right axis shows the total precipitation histogram.



### 4.3.1 Long-term simulation

The long-term effects of natural convection on sediment temperatures at the Babylon talus slope were examined through multi-
year simulations, focusing on an average hydrological year from the period of 2005 to 2020, where the MAAT was -1.4 °C
and the average maximum snow depth was 1.6 m. Consequently, the hydrological year 2009-2010 was chosen due to its air
temperatures and snow depth closely aligning with the period's average (MAAT: -1.3 °C; Max snow depth: 1.8 m). A spin-up
simulation was performed to establish a stable thermal regime by repeating the meteorological forcings for the 2009-2010
hydrological year across 100 cycles (Ross et al., 2021).

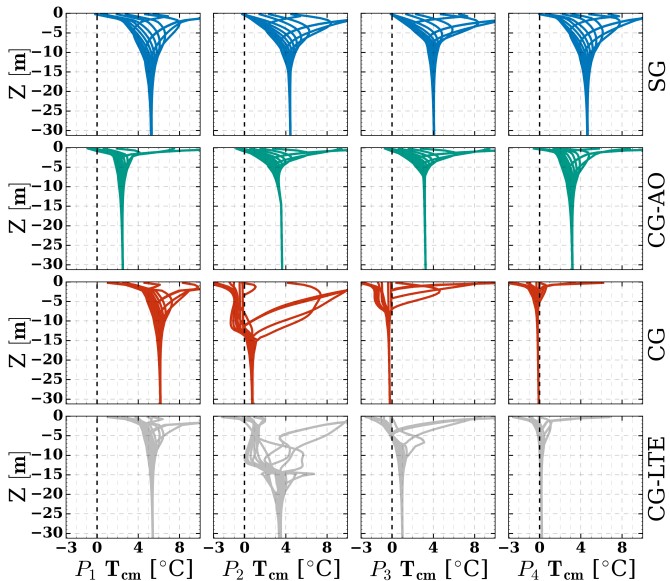

**Figure 11.** Sediment temperature profiles for the last day of each month for the last simulated year. The blue lines show the temperature
profiles for the S-GEOtop case (SG), the green lines show the temperature profiles for the CE-GEOtop case with the air convection module
turned off (CG-AO), the red lines show the temperature profiles for the CE-GEOtop case (CG), and the grey lines show the temperature
profiles for the CE-GEOtop case assuming LTE (CG-LTE). Profiles at $P_1$, $P_2$, $P_3$, and $P_4$ correspond to the upper, middle, and bottom
sections of the talus slope (Fig. 5C).

The analysis considered four cases using the same permeability of $k = 1.5 \times 10^{-6}$ m$^2$: (i) S-GEOtop (SG), (ii) CE-GEOtop
and the air convection module turned off (CG-AO), (iii) CE-GEOtop (CG), and (iv) CE-GEOtop assuming local thermal
equilibrium conditions (CG-LTE). Figure 11 shows the sediment temperature profiles for the last day of each month for the last
(i.e., 100th) simulated year. For all cases, bedrock is assumed to be almost impermeable with a $k = 2 \times 10^{-15}$ m$^2$ (Hayashi,
2020).

The SG case results indicate no permafrost in the Babylon talus slope and that temperatures remain relatively consistent
across all profiles. Surface temperatures drop below freezing for almost half the year, while temperatures at greater depths
stabilize around 5 °C. Similarly, the CG-AO results show no indication of permafrost. A comparison of the SG and CG-AO





results shows the effect of CE-GEOtop's shadow calculation and thermal conductivity modifications. The modified shadow computations result in lower temperatures in the upper profiles ($P_1$ and $P_2$) in the CG-AO case, as the upper zone of the
talus is closer to the rockwall, reducing incoming solar radiation. Conversely, the narrower temperature range at greater depths in CG-AO is due to the changes in thermal conductivity estimations, where a lower thermal conductivity leads to increased surface temperature gradients but reduces temperature changes with depth. These simulations, when compared with observed permafrost (Fig. 3), suggest that current climate conditions might not sustain permafrost, indicating permafrost thawing and initial formation under cooler climatic conditions.

The CG simulation, however, reveals a different situation and predicts the existence of permafrost at depths ranging from 10 to 35 m in the middle-lower section of the talus slope. Profiles $P_1$, $P_2$, and $P_3$ exhibit similar temperatures, whereas the upper profile $P_4$ indicates warmer conditions. These results indicate that the current climate conditions are capable of maintaining permafrost, with natural convection playing a critical role in preserving cold temperatures and permafrost in coarse sediments. A comparison of the CG and CG-AO results indicates that natural convection leads to a 3-4 °C reduction in temperatures in
the lower profiles ($P_3$ and $P_4$).

In the LTE simulation (CG-LTE), temperature profiles closely resemble those from the LTNE simulation (CG) at the extremities of the talus slope ($P_1$ and $P_4$). However, differences arise in the midsection ($P_2$ and $P_3$), where the LTNE approach indicates lower temperatures between 15 to 35 m. Conversely, the LTE approach reduces the cooling effect of natural convection, failing to predict permafrost. Natural convection's effectiveness in sediment cooling relies not only on air flux initiation
but also on the sinking of colder, denser air, which facilitates energy transfer throughout the sediment profile. This dynamic is overlooked in the LTE approach, where cold air cannot descend, restricting energy transfer from the surface to deeper depths. This phenomenon is evident in the temperature profiles at $P_2$ and $P_3$, where both simulations exhibit negative temperatures within the top 5 m of sediment. However, only the LTNE simulation shows deep-reaching cold temperature effects.

### 4.4 Permafrost effect on discharges

This section examines the effect of permafrost on spring discharges and temperatures by comparing the results of CG and CG-AO cases. The hydrological year 2007-2008 was simulated because measured discharge and temperature data of the Babylon basin were available for July-September 2008 (Muir et al., 2011). Figure 12 shows the summer 2008 simulated discharge and spring temperature. Since the GEOtop domain is smaller than the Babylon basin, the discharges are normalized by the area and shown in $\mathrm{mm/d}$. For profiles $P_3$ and $P_4$, sediment temperature, liquid water content, and ice content for four instants during
this period are shown in Appendix B (Figure B1).

The simulations for the scenario involving permafrost (CG) showed that the onset of spring flow occurred approximately two weeks later compared to the scenario without permafrost (CG-AO). In early May, the CG case exhibited negative temperatures and low water content throughout the talus, impeding infiltration and groundwater movement. On the other hand, the CG-AO scenario had positive temperatures, enabling groundwater to reach the base of the talus, increasing water content in this area
and resulting in earlier discharge at the model outlet.





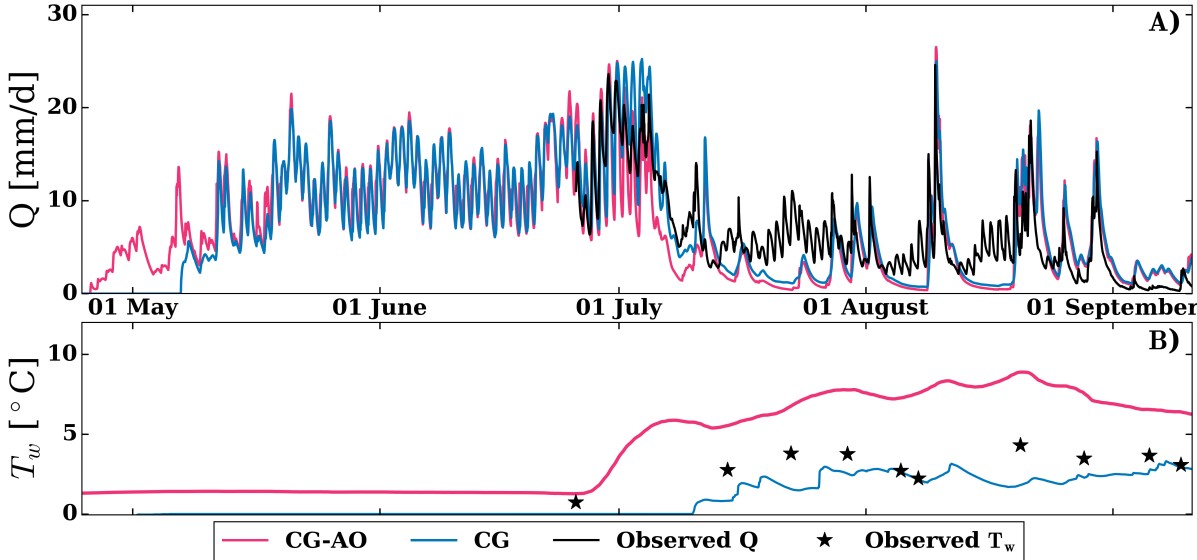

**Figure 12.** Effects of permafrost on discharge rates and spring water temperatures. (A) The simulated and observed discharges. (B) Simulated and observed temperatures.

From mid-May to July, both cases displayed similar behavior. However, in the CG case, slightly negative temperatures persisted in the middle-bottom of the talus, causing a minor delay in groundwater flow due to some fraction being frozen and retained. This behavior explains the minor differences in discharges during May and June between both cases. In the case with permafrost, discharge minimums tended to be higher, and the discharge maximums lower, suggesting a slightly slower response to snowmelt and rain due to water freezing and a slower hydraulic conductivity indicated by the negative temperatures and ice contents for profiles $P_3$ and $P_4$ during this period.

The discharge values in both cases align with observations in late June and early July. However, starting in mid-July, the snowmelt contribution from late-lying snowpack (present in the upper part of the Babylon talus slope as depicted in Fig. 3) in zones not considered in the model became more significant. As a result, the simulated discharge showed lower magnitudes than the observed values, matching the observed discharges only during precipitation events. The case with permafrost showed better agreement with the observed discharges than the case without permafrost during late June and early July, indicating a more accurate representation of the basin discharge. Additionally, when comparing the spring temperatures to the measured data, it's evident that the temperatures from the model with permafrost closely resemble the measured values, while the case without permafrost overestimates the spring temperatures. These findings support the idea that the low water temperature during the summer may be related to the presence of coarse sediments and permafrost occurrence.



## 5   Conclusions

This study improved the GEOtop 3.0 model (S-GEOtop) by incorporating natural convection through the local thermal non-equilibrium (LTNE) approach. The LTNE approach accounts for a different temperature for the air inside the sediment pores with respect to the other phases, including water, ice, and sediments. Additionally, adjustments were made to the thermal
conductivity calculations and cast shadows to enhance the energy simulations in alpine landforms. The modified convection-enhanced GEOtop (CE-GEOtop) model was evaluated on a talus slope in the Canadian Rockies with permafrost identification from field data.

Talus slopes with coarse sediments exhibit a thermal anomaly, where the middle-lower sections present lower temperatures than the top of the talus. The CE-GEOtop model successfully captured this phenomenon, depicting an inflow of cold air at the
base of the talus slope and warmer air exiting at the top during winter months, as well as a reversal of this flow during summer months. The strength of natural convection is defined by the Darcy-Rayleigh number ($Ra$). Simulations indicated that flow initiates when $Ra$ exceeds $Ra_c$, but larger $Ra$ is necessary for natural convection impacting sediment temperatures. Moreover, at larger timescales, sporadic instances with large $Ra$ are insufficient for inducing cooling, necessitating either recurrent or prolonged periods with elevated temperature gradients.

The simulations showed that air temperature within the sediments rarely reaches local thermal equilibrium with the other phases in the presence of natural convection. As a result, the long-term simulations performed using the local thermal equilibrium approach overestimated temperatures and failed to predict permafrost presence in the talus slope, underscoring the significance of considering thermal non-equilibrium conditions.

The S-GEOtop simulations under current climate conditions also failed to predict permafrost in the talus, but the CE-GEOtop
model successfully predicted permafrost from depths of 10 to 35 m in the middle-bottom section of the slope. This highlights the important role of natural convection in coarse sediments' energy balance and the relevance of using the LTNE approach to represent this process correctly. The presence of permafrost affects the hydrological response of the talus, resulting in delayed discharges after snowmelt and rain due to partial groundwater freezing in the sediment's lower layer. During summer, persistently cold discharges were simulated when permafrost was present, matching the measured temperatures at the talus outlet.
Overall, this study demonstrated the important role of natural convection in permafrost dynamics within alpine landforms, as well as the necessity to consider this physical process in heat transport modeling in cold environments dominated by coarse blocky sediments.

CE-GEOtop offers comprehensive capabilities to simulate snow cover, ground temperatures, permafrost, and water budgets, providing a new framework to analyze the interplay between groundwater discharge and permafrost, as well as the effects
of surficial geology on temperature distribution. Future research should address the interplay between air temperatures, snow depth, and natural convection under a warmer climate, particularly to forecast sediment temperature, permafrost extent, and groundwater discharges. Additionally, further examination of the effect of surficial geology on permafrost distribution and natural convection is necessary, with particular attention given to the role of spatial heterogeneities in permeability, deposit





depth, and bedrock configuration. All these factors strongly influence the strength of natural convection and the patterns of airflow within the sediments.

*Code availability.*  CE-GEOtop is provided with a GNU General Public License, version 3 (GPL-3.0). The source code and a test case are available through GitHub at the address: https://github.com/gzegers/geotop-CE

*Author contributions.*  GZ and MH conceived and designed the analysis; GZ, MH, and RP contributed to the methodology and analysis; GZ wrote the code; GZ performed the analysis, analyzed the results, and made the figures; GZ, MH, and RP contributed to writing, reviewing, and editing.

*Competing interests.*  The authors declare no potential competing interests.

*Acknowledgements.*  This study was funded by the Natural Sciences and Engineering Research Council of Canada (Discovery Grant), the Canada First Research Excellence Fund (Global Water Futures), and Alberta Innovates (Water Innovation Program). GZ acknowledges financial support from the National Research and Development Agency (ANID) Doctoral Grant 72200390. Additionally, OpenAI's GPT-4 language model (ChatGPT-4) was utilized for reviewing text spelling and grammar.



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

## Appendix A: Governing equations

### A1    Numerical implementation

The four main equations are solved in a time-lagged manner, following the following order: (i) air mass balance (Eq. 5), (ii) air
energy balance (Eq. 8), (iii) composite medium energy balance (Eq. 4) and, (iv) water mass balance (Eq. 3). These equations
can be generalized using the following form:

$$\frac{\partial F(\chi)}{\partial t} + \nabla \cdot (-L(\chi)\nabla\chi) + S = 0 \tag{A1}$$

Here, $\chi$ represents the unknown variable function of space and time, $F$ is a non-linear function, $S$ is the sink term, and $L$ is
a property dependent on $\chi$. By integrating this equation over a control volume $V_c$ and using an implicit approach, we obtain:

$$\frac{(F(\chi_i^{n+1}) - F(\chi_i^n))\Omega\Delta z}{\Delta t} - \sum_j^M \frac{L_{ij}^{n+1}\Omega_{ij}}{D_{ij}}(\chi_j^{n+1} - \chi_i^{n+1}) \tag{A2}$$

$$+ S_w\Omega\Delta z = G_{il}(\chi^{n+1}) \approx 0$$

This equation is written for the generic $i$th cell, where $n$ represents the previous time step at which the solution is known,
and $n+1$ is the next time step at which the solution is unknown. $\Delta t$ is the time step, $j$ is the index of the m adjacent cells with
which the $i$th cell can exchange fluxes, $k_{ij}$ represents the conductivity between cell $i$ and $j$, $D_{ij}$ denotes the distance between
the centers of cells $i$ and $j$, $S_i$ represents the sink terms, and $G_i$ is the residual to be minimized to find a solution. As explained
in Endrizzi et al. (2014), Eq. A2 is solved using the Newton-Rhapson method.





## A2  Mass balance

The governing coupled equation for two-phase flow is the following (Nield and Bejan, 2017):

$$
\frac{\partial(\rho_w\theta_w + \rho_i\theta_i)}{\partial t} + \frac{\partial \rho_a\theta_a}{\partial t} + \rho_w\nabla\cdot(\boldsymbol{J}_w) + \rho_a\nabla\cdot(\boldsymbol{J}_a) + S_g + S_w = 0 \tag{A3}
$$


To solve this system, we will not consider the interplay between the fluxes of water and air. This means that the movement of water will not cause air movement, and air movement will not cause water movement. With this assumption, we can separate Eq. A3 by using the split method, getting:

$$
\frac{\partial(\rho_w\theta_w + \rho_i\theta_i)}{\partial t} + \rho_w\nabla\cdot(\boldsymbol{J}_w) + S_w = 0 \tag{A4}
$$

$$
\frac{\partial(\rho_a\theta_a)}{\partial t} + \rho_a\nabla\cdot(\boldsymbol{J}_a) + S_a = 0 \tag{A5}
$$


Eq. A4 is used to solve the water flow and Eq. A5 to solve the air flow. Using the Oberbeck–Boussinesq approximation to solve the airflow, the air density ($\rho_a$) can go out of the partial time derivative. The air content in a control volume can only change if there is a change in water or ice content($\partial\theta_a = -\partial\theta_w - \partial\theta_i$), but since we neglected the interplay between the fluxes of water and air, we get that $\frac{\partial(\theta_a)}{\partial t} \sim 0$. This assumption is valid as, in this case, temperature gradients dominate the air fluxes and are not by the forces generated by the water movement. Now, we get to the final equation for the airflow:


$$
\rho_a\nabla\cdot(\boldsymbol{J}_a) + S_g = 0 \tag{A6}
$$

This equation is integrated over a volume $V_c$, and considering that $J_a = -\frac{k_{ra}k}{\mu_a}(\nabla P - \rho_a\mathbf{g})$, this yield:

$$
-\sum_{j}^{M} \frac{k_{ija}^{n+1}\Omega_{ij}}{D_{ij}}(P_{jg}^{n+1} - P_{ia}^{n+1}) - \frac{K_{ija}^{n+1}\Omega_{ij}}{D_{ij}}Tb_{ia}^{n+1}(z_{jg} - z_{ia})\bigg|_{top} + \tag{A7}
$$
$$
-\frac{K_{ija}^{n+1}\Omega_{ij}}{D_{ij}}Tb_{ia}^{n+1}(z_{jg} - z_{ia})\bigg|_{bottom} + S_g\Omega\Delta z = G_{ia}(P_{ia}^{n+1}) \sim 0,
$$

$Tb_i^{n+1} = \rho_{rg}g(1 - \beta(T_{ia}^{n+1} - T_{ref}))$ represents the buoyancy term and $K_a = \frac{k_{ra}k}{\mu_a}$ the air conductivity.

## A3  Energy balance

The heat transport equation to solve for the air energy balance was presented in Eq. 8. Similarly, as before, this equation is integrated over a volume $V_c$,





$$\frac{\Delta z \rho_a C_a \Omega}{\Delta t}(\theta_{ia}^{n+1}(T_{ia}^{n+1} - T_{ia}^n)) - \sum_{j}^{M} \frac{\theta_{gij}\lambda_g \Omega_{ij}}{D_{ij}}(T_{jg}^{n+1} - T_{ia}^{n+1})+ \tag{A8}$$

$$\sum_{j}^{M} (\boldsymbol{J_{ijg}^n}\Omega_{ij}\rho_a C_a(T_{ijg}^{n+1} - T_{ref}))\bigg|_{face} + \Delta z \Omega h T_{ia}^{n+1}+ \tag{A9}$$

$$- \Delta z \Omega h T_{icm}^{n+1} + S_g \Omega \Delta z = RG_{ia}(T_{ia}^{n+1}) \sim 0$$

## Appendix B: Additional Figures



**Figure B1.** Effects of permafrost on discharge rates and spring water temperatures. Blue lines represent the results for the CG case, while pink lines correspond to the CG-AO case. $A_1$), $A_2$), and $A_3$) depict the simulated sediment temperatures ($S_T$), liquid water content ($\theta_{liq}$), and ice content $\theta_{ice}$) for profile $P_3$ at four distinct times during the summer of 2008, respectively. In these panels, the fine sediment layer is indicated with a brown color, and the bedrock is indicated with a grey color. Panel B) compares the simulated discharges with the observed discharges (represented by the black line), and Panel C) contrasts the simulated temperatures with the observed temperatures (indicated by black stars). The green dashed vertical lines in panels B) and C) indicate the four distinct times where the profiles $P_3$ and $P_4$ are shown. Panels $D_1$), $D_2$), and $D_3$) display the simulated sediment temperatures ($S_T$), liquid water content ($\theta_{liq}$), and ice content ($\theta_{ice}$) for profile $P_53$ at four distinct times during the summer of 2008, respectively. In these panels, the fine sediment layer is indicated with a brown color, and the bedrock is indicated with a grey color.