# Peer review of "Improved permafrost modeling in mountain environments by including air convection in a hydrological model"

_EGUsphere, 2024_

## Referee Comment (RC2)

Review of G. Zegers et al. (2024)
https://doi.org/10.5194/egusphere-2024-2575

G. Zegers and colleagues enhanced GEOtop-3D, a well-known physically-based distributed hydrological model, to include the thermal effects of air convection and local thermal nonequilibrium (LTNE). Air convection exerts a cooling effect and critically shapes the permafrost distribution at landform scale, the scale needed for hydrological modelling in complex high-mountain catchments. The freely available model is (to our knowledge) the first geoscientific model that includes two-phase energy balance equations, four phases (rock-ice-water-air), and an explicit treatment of air and water flow. This is a valuable and important contribution and presented together with a very concisely written publication. The modelling is well connected to field observations and to the geomorphological literature where undercooling has been known for centuries.

We have only few points to criticize. A major point is that the authors did not split in their paper results and discussion. This makes it difficult for the reader to see the results clearly separated from the interpretations. This is particularly true for Chapter 4.4 where the authors suddenly discuss the hydrological implications of permafrost on discharge. It would be nice to have the hydrological outcomes of this paper much better included in the whole paper, starting in the introduction and then also in the results and discussion section.  We will give more details below (referring to L439).

Another point is that below-ground heat transfer by thermal radiation is ignored, which can be significant for cobble to boulder-sized rocks without fine sediments that would clog the large pore space. This has been shown in geotechnical experiments by Fillion et al. (2011) ("Radiation heat transfer becomes significant for $d_{10}$ higher than 10 mm and predominant at values higher than 90 mm") and in the field by Scherler et al. (2014) and Amschwand et al. (2024). Radiation increases the stagnant thermal conductivity of the solid composite material. It does probably not change your main findings significantly, though, because the effective thermal conductivity values of ~1 W m$^{-1}$ K$^{-1}$ would be again closer to those of the standard GEOtop implementation. Nonetheless, because large pores for radiation and high permeability for strong convection are likely to occur together in rocky debris, the issue should be briefly discussed (e.g., in the Section at L191–218).

A few minor points are listed below.

Minor points

L13: Please consider rephrasing to "…local thermal equilibrium approaches underestimate the impact of natural convection especially on short timescales."

L28: Please change the reference from Riseborough et al. 2008 to a more specific reference like Van Everdingen, R. O., 1998.

L28/29: …'positive mean annual air temperature' this information was already explored far before the year 2000 as it is shown in your references. Already in the 1930s local researchers in Switzerland and elsewhere (e.g. Bächler 1930s) found such effects at certain places and in the early 1990s this was already scientifically described (e.g. Hoelzle 1992).

L40: You may add: https://doi.org/10.1002/esp.5998 (Wiegand & Kneisel., 2024)

L59: Suitable reference to add (and maybe to discuss) is Wicky et al. (2024), who also modelled air convection in a talus slope with a similar approach. Furthermore, please mention the works by Marchenko (2001) (a two-phase LTNE, 1-dim model where Rayleigh convection is parameterized), and Tanaka et al. (2000).

L112: Please consider rephrasing: "...reducing the amount of energy gained by the ground from the atmosphere (Gruber and Hoelzle, 2008)".

L127: "$F_w$ as a sink term representing evapotranspiration": Please clarify by saying that $F_w$ affects the surface cells only (correct?) and is derived from the surface energy balance.

L165: You could consider Amschwand et al. (2024, Fig. 4) for the closure of the snowpack.

L170: Please consider rephrasing (see major point): "The energy transport equation in the air phase combines advection and conduction processes, but it does not account for the thermal radiation of coarse sediments". We suggest not to mention thermal radiation in the paragraph on the air energy equation (because air is essentially IR-transparent at such short distances), and to discuss it instead in the paragraph on the composite-medium conductivity (Sect. 2.3.2).

L278: In the description of the Babylon talus slope, please mention the typical block size.

L377: "...air temperature, incoming longwave radiation, and cloud fraction (estimated from incoming shortwave radiation data (Long et al., 2006))". It is not clear here how your input data is organized. Which radiation components have been measured, short-wave as well? Why did you use cloud fraction instead of the shortwave incoming radiation directly?

L439: While Chapter 4.3 follows naturally from the previous discussion (permafrost distribution), Chapter 4.4 about the effects on discharge appears a bit sudden. A statement at the beginning of Chapter 4.4 or in the introduction about the links between ground thermal regime, ice content, and hydraulic properties (permeability) would make the transition clearer. While the interpretation of water temperatures (Fig. 12B) follows straightforward from the previous discussions, the discharge (Fig. 12A) is hard to interpret solely with the concepts presented in this paper. For example, the discharge is flashier in the CG-AO (no permafrost) model run than in the CG (permafrost) model run ("in the case with permafrost, discharge minimums tended to be higher, and the discharge maximums lower,..."), which was attributed to refreezing and lower hydraulic permeability (L454). First: is this a conjecture or a model output? Second, often the opposite is true, supra-permafrost runoff being more rapid and flashy than runoff in a non-permafrost catchment. Do these findings refer to deeper intra-talus water pathways? Please clarify, perhaps explaining (or showing) the different water pathways in the frozen/non-frozen talus.

L597: The links for the Leontini reference do not work.

L699: Please briefly comment on: How strong is the influence of moisture on the air density, and would taking the virtual temperature (or even the virtual potential temperature) improve the modelling?

Figure 2: The figure is helpful. Please clarify that the CM energy balance is 1D vertical, or maybe draw 3D cubes instead of 2D squares.

Figure 3: It is not clear why you do not use your GEOtop model to do the shortwave incoming radiation calculation as you explain also in the text (L311). In addition, your BTS-class -

3°C<BTS<-2°C is not shown in the figure, but maybe there are no measurements within this BTS-class, please explain?

References

Amschwand, D., Scherler, M., Hoelzle, M., Krummenacher, B., Haberkorn, A., Kienholz, C., and Gubler, H.: Surface heat fluxes at coarse blocky Murtèl rock glacier (Engadine, eastern Swiss Alps), The Cryosphere, 18, 2103–2139, https://doi.org/10.5194/tc-18-2103-2024, 2024.

Bächler, E., 1930, Der verwünschte oder verhexte Wald im Brüeltobel: Appenzellerkalender, 209.

Fillion, M.-H., Côté, J., and Konrad, J.-M.: Thermal radiation and conduction properties of materials ranging from sand to rock-fill, Canadian Geotechnical Journal, 48, 532–542, https://doi.org/10.1139/t10-093, 2011.

Hoelzle, M., 1992, Permafrost occurrence from BTS measurements and climatic parameters in the eastern Swiss Alps: Permafrost and Periglacial Processes, v. 3, no. 2, p. 143-147.

Marchenko, S.: A model of permafrost formation and occurrences in the intracontinental mountains, Norsk Geografisk Tidsskrift - Norwegian Journal of Geography, 55:4, 230-234, DOI: 10.1080/00291950152746577, 2001.

Scherler, M., Schneider, S., Hoelzle, M., and Hauck, C.: A two-sided approach to estimate heat transfer processes within the active layer of the Murtèl–Corvatsch rock glacier, Earth Surf. Dynam., 2, 141–154, https://doi.org/10.5194/esurf-2-141-2014, 2014.

Tanaka, HL, Nohara, D., and Yokoi, D.: Numerical simulation of wind hole circulation and summertime ice formation at Ice Valley in Korea and Nakayama in Fukushima, Japan, Journal of the Meteorological Society of Japan. Ser. II 78.5, 611-630, 2000. https://www.jstage.jst.go.jp/article/jmsj1965/78/5/78_5_611/_pdf/-char/ja http://gpvjma.ccs.hpcc.jp/~tanaka/web/papers/tnb2006_pdf.pdf

Van Everdingen, R. O., 1998, Multi-Language Glossary of Permafrost and related Ground-Ice Terms, in Association, I. P., ed.: Calgary, University Printing Services of the University of Calgary.

Wicky, J., Hilbich, C., Delaloye, R. and Hauck, C. (2024), Modeling the Link Between Air Convection and the Occurrence of Short-Term Permafrost in a Low-Altitude Cold Talus Slope. Permafrost and Periglac Process, 35: 202-217. https://doi.org/10.1002/ppp.2224

---

## Author Comment (AC2)

We thank the reviewers for their time in reviewing our paper. We provide responses to each individual point below. For clarity, comments are given in italics, and our responses are given in plain blue text.

*G. Zegers and colleagues enhanced GEOtop-3D, a well-known physically-based distributed hydrological model, to include the thermal effects of air convection and local thermal nonequilibrium (LTNE). Air convection exerts a cooling effect and critically shapes the permafrost distribution at landform scale, the scale needed for hydrological modelling in complex high-mountain catchments. The freely available model is (to our knowledge) the first geoscientific model that includes two-phase energy balance equations, four phases (rock-ice- water-air), and an explicit treatment of air and water flow. This is a valuable and important contribution and presented together with a very concisely written publication. The modelling is well connected to field observations and to the geomorphological literature where undercooling has been known for centuries. We have only few points to criticize.*

We appreciate the supportive comments.

*A major point is that the authors did not split in their paper results and discussion. This makes it difficult for the reader to see the results clearly separated from the interpretations. This is particularly true for Chapter 4.4 where the authors suddenly discuss the hydrological implications of permafrost on discharge. It would be nice to have the hydrological outcomes of this paper much better included in the whole paper, starting in the introduction and then also in the results and discussion section. We will give more details below (referring to L439).*

We will address this concern in the detailed responses below and will add a paragraph to the Introduction to better introduce the hydrological aspects. We will also revise Section 4.4 to include additional detail and improve clarity in the interpretation of the results.

*Another point is that below-ground heat transfer by thermal radiation is ignored, which can be significant for cobble to boulder-sized rocks without fine sediments that would clog the large pore space. This has been shown in geotechnical experiments by Fillion et al. (2011) ("Radiation heat transfer becomes significant for d10 higher than 10 mm and predominant at values higher than 90 mm") and in the field by Scherler et al. (2014) and Amschwand et al. (2024). Radiation increases the stagnant thermal conductivity of the solid composite material. It does probably not change your main findings significantly, though, because the effective thermal conductivity values of ~1 W m71 K-1 would be again closer to those of the standard GEOtop implementation. Nonetheless, because large pores for radiation and high permeability for strong convection are likely to occur together in rocky debris, the issue should be briefly discussed (e.g ., in the Section at L191-218).*

We agree that incorporating thermal radiation could enhance the representation of the energy balance. However, it is beyond the scope of the present work. Accounting for thermal radiation in the current model setup is particularly challenging due to the presence of four distinct phases

(sediment, air, water, and ice), all of which need to be considered when estimating radiative heat transfer. We will include a brief discussion of thermal radiation in Section 2.3.2, where thermal conductivity calculations are introduced.

*A few minor points are listed below.*

*Minor points L13: Please consider rephrasing to " ... local thermal equilibrium approaches underestimate the impact of natural convection especially on short timescales."*

We will accept the suggestion and rephase L13 to " ... local thermal equilibrium approaches underestimate the impact of natural convection especially on short timescales."

*L28: Please change the reference from Riseborough et al. 2008 to a more specific reference like Van Everdingen, R. O ., 1998.*

We agree, we will change the reference to Van Everdingen, R. O ., 1998.

*L28/29: ... 'positive mean annual air temperature' this information was already explored far before the year 2000 as it is shown in your references. Already in the 1930s local researchers in Switzerland and elsewhere (e.g. Bächler 1930s) found such effects at certain places and in the early 1990s this was already scientifically described (e.g. Hoelzle 1992).*

Thanks for the suggestion. We will include those references in the revised manuscript.

*L40: You may add: https://doi.org/10.1002/esp.5998 (Wiegand & Kneisel ., 2024)*

We will include the suggested reference.

*L59: Suitable reference to add (and maybe to discuss) is Wicky et al. (2024), who also modelled air convection in a talus slope with a similar approach. Furthermore, please mention the works by Marchenko (2001) (a two-phase LTNE, 1-dim model where Rayleigh convection is parameterized), and Tanaka et al. (2000).*

We will mention Wicky et al. (2024), Marchenko (2001), and Tanaka et al. (2000) in the revised manuscript.

*L112: Please consider rephrasing: " ... reducing the amount of energy gained by the ground from the atmosphere (Gruber and Hoelzle, 2008)"*

We will rephrase to: " ... reducing the amount of energy gained by the ground from the atmosphere (Gruber and Hoelzle, 2008)"

*L127: "Fw as a sink term representing evapotranspiration": Please clarify by saying that Fw affects the surface cells only (correct?) and is derived from the surface energy balance.*

Fw can act in the first layers of the sediments. Particularly transpiration can go into deeper layers of the sediments depending on the model configuration. We will clarify that in the revised manuscript.

*L165: You could consider Amschwand et al. (2024, Fig. 4) for the closure of the snowpack.*

We will include Amschwand et al. (2024) in the mentioned references.

*L170: Please consider rephrasing (see major point): "The energy transport equation in the air phase combines advection and conduction processes, but it does not account for the thermal radiation of coarse sediments". We suggest not to mention thermal radiation in the paragraph on the air energy equation (because air is essentially IR-transparent at such short distances), and to discuss it instead in the paragraph on the composite-medium conductivity (Sect. 2.3.2).*

We agree with the suggestion. We will mention thermal radiation in the paragraph on the composite-medium conductivity

*L278: In the description of the Babylon talus slope, please mention the typical block size.*

We will include in the description that:

"The talus is predominantly composed of cobble to boulder-sized rocks and lacks fine sediments"

*L377: " ... air temperature, incoming longwave radiation, and cloud fraction (estimated from incoming shortwave radiation data (Long et al ., 2006))". It is not clear here how your input data is organized. Which radiation components have been measured, short-wave as well? Why did you use cloud fraction instead of the shortwave incoming radiation directly?*

Because incoming solar radiation is not constant in the domain and it depends on the shadow effect (Section 2.3.4), measured income solar radiation cannot be used directly. Instead, incoming solar radiation is used to estimate hourly cloud fraction and estimate incoming solar radiation to each pixel of the model based on the hourly sun position and estimated cloud fraction. This will be explained in the revised manuscript.

*L439: While Chapter 4.3 follows naturally from the previous discussion (permafrost distribution), Chapter 4.4 about the effects on discharge appears a bit sudden. A statement at the beginning of Chapter 4.4 or in the introduction about the links between ground thermal regime, ice content, and hydraulic properties (permeability) would make the transition clearer.*

We will include a paragraph in the introduction referring to permafrost and subsurface water flow:

"The presence of permafrost directly influences subsurface water flow. As soil reaches freezing temperatures, water becomes ice, hindering or blocking flow depending on the ice content. Permafrost generally acts as a confining layer, creating two main flow paths: near-surface flow above the permafrost (supra-permafrost) and deeper subsurface flow (sub-permafrost) (Giardino et al., 1992; Harrington et al., 2018; Langston et al., 2011; Woo et al., 1994. However, when permafrost is discontinuous or contains low ice content, intra-permafrost flow may also occur. Regarding the contribution of ice melt to streamflow, several studies have shown it is not highly significant in alpine landforms (Arenson and Jakob, 2010; Harrington et al., 2018; Hayashi, 2020;

Krainer and Mostler, 2002). Thus, the primary function of permafrost in alpine hydrogeology is to control the subsurface water flow paths."

*While the interpretation of water temperatures (Fig. 12B) follows straightforward from the previous discussions, the discharge (Fig. 12A) is hard to interpret solely with the concepts presented in this paper. For example, the discharge is flashier in the CG-AO (no permafrost) model run than in the CG (permafrost) model run ("in the case with permafrost, discharge minimums tended to be higher, and the discharge maximums lower ,... "), which was attributed to refreezing and lower hydraulic permeability (L454). First: is this a conjecture or a model output?*

*Do these findings refer to deeper intra-talus water pathways? Please clarify, perhaps explaining (or showing) the different water pathways in the frozen/non-frozen talus.*

*Second, often the opposite is true, supra-permafrost runoff being more rapid and flashy than runoff in a non-permafrost catchment.*

This conclusion is based on an interpretation of the model results. The findings specifically refer to intra-talus water pathways. In the analyzed case, there is no supra-permafrost layer because the simulated permafrost is discontinuous with low ice content rather than a continuous body. As a result, there is no supra-permafrost flow. Instead, groundwater within the talus flows from more conductive layers toward less conductive but more saturated zones near the bedrock.

During June for the CG case, the snowmelt and infiltration led to an increase in ice content within the lower conductivity layer at the middle section of the talus (P3 profile). This increase is illustrated in the following figure (provided in the Appendix).

We will clarify in the text that we are referring to intra-talus flow.

[Figure]

*L597: The links for the Leontini reference do not work.*

We apologize for the error in the original reference. The correct reference is "Su, Y., & Davidson, J. H. (2015). Modeling approaches to natural convection in porous media. Springer." https://link.springer.com/book/10.1007/978-3-319-14237-1

*L699: Please briefly comment on: How strong is the influence of moisture on the air density, and would taking the virtual temperature (or even the virtual potential temperature) improve the modelling?*

Moisture in the air has a minimal effect on air density at temperatures below 20 °C; therefore, including it would not significantly impact the simulation results. This will be clarified in the revised manuscript.

*Figure 2: The figure is helpful. Please clarify that the CM energy balance is 1D vertical, or maybe draw 3D cubes instead of 2D squares.*

Yes, the CM energy balance is 1D vertical. We will change the figures to 3D cubes to show a clearer description.

*Figure 3: It is not clear why you do not use your GEOtop model to do the shortwave incoming radiation calculation as you explain also in the text (L311).*

You are right that the radiation calculation can be done directly with the GEOtop model however, the PISR values in this figure are shown as a qualitative analysis and cover a larger extent than one the employed GEOtop model. Nonetheless, the radiation calculations done by using SAGA GIS do consider the shadow effect in incoming radiation.

*In addition, your BTS-class - 3°C<BTS <- 2°C is not shown in the figure, but maybe there are no measurements within this BTS- class, please explain?*

Yes, there are no measurements within the BTS-class - 3°C<BTS <- 2°C.

*References*

*Amschwand, D., Scherler, M., Hoelzle, M., Krummenacher, B., Haberkorn, A., Kienholz, C., and Gubler, H.: Surface heat fluxes at coarse blocky Murtèl rock glacier (Engadine, eastern Swiss Alps), The Cryosphere, 18, 2103-2139, https://doi.org/10.5194/tc-18-2103-2024, 2024.*

*Bächler, E., 1930, Der verwünschte oder verhexte Wald im Brüeltobel: Appenzellerkalender, 209.*

*Fillion, M.-H., Côté, J., and Konrad, J.-M.: Thermal radiation and conduction properties of materials ranging from sand to rock-fill, Canadian Geotechnical Journal, 48, 532-542, https://doi.org/10.1139/t10-093, 2011.*

*Hoelzle, M., 1992, Permafrost occurrence from BTS measurements and climatic parameters in the eastern Swiss Alps: Permafrost and Periglacial Processes, v. 3, no. 2, p. 143-147.*

*Marchenko, S.: A model of permafrost formation and occurrences in the intracontinental mountains, Norsk Geografisk Tidsskrift - Norwegian Journal of Geography, 55:4, 230-234, DOI: 10.1080/00291950152746577, 2001.*

*Scherler, M., Schneider, S., Hoelzle, M., and Hauck, C.: A two-sided approach to estimate heat transfer processes within the active layer of the Murtel-Corvatsch rock glacier, Earth Surf. Dynam., 2, 141-154, https://doi.org/10.5194/esurf-2-141-2014, 2014.*

*Tanaka, HL, Nohara, D., and Yokoi, D.: Numerical simulation of wind hole circulation and summertime ice formation at Ice Valley in Korea and Nakayama in Fukushima, Japan, Journal of the Meteorological Society of Japan. Ser. Il 78.5, 611-630, 2000. https://www.jstage.jst.go.jp/article/jmsj1965/78/5/78_5_611/_pdf/-char/ja http://gpvjma.ccs.hpcc.jp/~tanaka/web/papers/tnb2006_pdf.pdf*

*Van Everdingen, R. O., 1998, Multi-Language Glossary of Permafrost and related Ground-Ice Terms, in Association, I. P., ed.: Calgary, University Printing Services of the University of Calgary.*

*Wicky, J., Hilbich, C., Delaloye, R., and Hauck, C. (2024), Modeling the Link Between Air Convection and the Occurrence of Short-Term Permafrost in a Low-Altitude Cold Talus Slope. Permafrost and Periglac Process, 35: 202-217. https://doi.org/10.1002/ppp.2224*

---

## Author Response (AR1)

We thank the anonymous reviewer for their time in reviewing our paper. We provide responses to each individual point below. For clarity, comments are given in italics, and our responses are given in plain blue text.

*Permafrost and its reaction on climate change are key issues of cryosphere research and thus also in TC. As the processes that change permafrost are very complex, process understanding is still comparatively limited. Improving the modelling of permafrost is an important approach to close this research gap. The work of Zegers et al. addresses relevant scientific questions in the context of permafrost research by improving its modelling through introduction of air convection and its effects (heat advection-conduction without assuming a local thermal equilibrium between the air and matrix) into the 3D hydrology model GEOtop. In this way, new methods and results for permafrost modelling based on the existing theory of air convection and density-driven buoyancy flow of air and heat advection conduction in a porous matrix are presented.*

*I found the paper very interesting, generally well written, well structured and easy to follow. Only the large number of acronyms sometimes made it difficult to follow the storyline. The title of the paper is well chosen and reflects the content well. The abstract is short and concise and contains key information of the paper. Due to the topic of the study, the paper contains a larger number of mathematical formulae, but these are balanced in scope, well presented and explained in the text as well as are useful for understanding the content. Results are clearly sufficient to support the interpretations and conclusions, which are clear and useful for the research community. In general, the paper is well embedded in appropriate international references (see also my minor comment below).*

*I suggest only minor revision before the paper can be accepted in the Cryosphere.*

We appreciate the supportive comments.

*Minor comments:*

*Supercooled talus slopes are an interesting phenomenon in the context of permafrost processes. This is well described and introduced in the paper, but I suggest to mentioned that supercooled talus slopes have been known for a long time (e.g. since the end of the 18th century).*

We included a sentence of the introduction referring to that (lines 43-45 marked manuscript):

"Abnormally low ground temperatures at relatively low elevations, often indicative of isolated permafrost patches, have long been documented in porous debris accumulations such as talus slopes and, in some cases, relict rock glaciers (Morard et al., 2010; Sawada et al., 2003; Wakonigg, 1996).

*I also wondered to what extent the distinction between summer and winter situations for air circulation in the talus slopes is too general. In principle, the difference between day and night often has a similar effect, especially if, for example, long-wave radiation at the surface (radiation night) causes a strong cooling of the upper talus sediment layers near the surface at night. Then a corresponding density gradient of the air in the talus body should also occur. Of course, the time available to drive this process is very limited, but at least it can counteract the daytime situation in summer. Perhaps this process is taken into account in the paper, but I couldn't find it in the text.*

The distinction between summer and winter air circulation regimes in the manuscript is intended to illustrate the model's capability to reproduce the dominant seasonal patterns of convective airflow in talus slopes, where the direction of flow and its thermal impacts are more clearly defined. We agree that similar convective mechanisms can also occur on shorter timescales, such as the diurnal cycle. To clarify this point, we have included the following text (lines 349-352 marked manuscript):

" Similar convective patterns can also occur on a diurnal timescale: surface cooling by longwave radiation at night can drive upslope air movement, while daytime heating by shortwave radiation can reverse this flow as surface temperatures rise."

While these short-term processes are not explicitly analyzed in the summer and winter regime simulations, they are indeed accounted for in the developed model through the full surface energy balance. This effect is taken into account in the long-term simulations that are forced with half-hourly meteorological data.

*It also appears that the winter and summer simulations shown are driven by constant temperature values and do not take into account daily temperature variations.*

Yes, constant external air temperatures were used in the summer and winter simulations, which were designed as simplified test cases to isolate and illustrate the direct effects of higher or lower air temperatures on airflow patterns. This setup was intended to demonstrate the model's behavior under contrasting thermal conditions. However, this is not a limitation of the model itself; it is fully capable of incorporating time-varying boundary conditions. We have rephrased this (lines 370-375 marked manuscript):

"Although CE-GeoTOP is capable of incorporating fully time-varying meteorological forcings, the summer and winter regime simulations were conducted with constant boundary conditions for 10 days to isolate the direct effects of higher or lower air temperatures on airflow patterns. As in the preceding experiments, all surface energy-balance terms were neglected."

*In the long-term simulation in Chapter 4.3.1, it is also not clear at which temporal resolution the driving data (e.g. of air temprature, radiation etc. ) are used. This should be clarified.*

For the long-term simulations the model was forced with half-hourly meteorological data from the nearby (~0.5 km) Opabin weather station. This is mentioned in line 413 of the marked manuscript.

*I am not familiar with the study area presented, but it seems to me that there are simpler situations with supercooled talus slopes (e.g. in the European Alps) where the temperature effects are measurable (and have actually been measured). These conditions would be much more suitable for evaluating the model, as the temperature could be evaluated directly (and air convection could actually be measured). In the supercooled talus slopes of the Alps, the temperature effect is also very clearly visible in the vegetation. Is there such evidence in the vegetation of the study area in Canada?*

In this study, we chose to focus on permafrost dynamics in the Canadian Rockies, an area that has received less attention in the literature compared to regions like the European Alps. We agree that the supercooled talus slopes of the Alps, with their more available data, would provide an excellent context for further evaluation of the model.

Regarding the use of vegetation as an indicator of ground temperature, we note that the study site in the Canadian Rockies is predominantly composed of coarse, blocky sediments with very limited vegetation cover. As a result, it is difficult to delineate thermally anomalous zones based on vegetation patterns. We included the following phrase to indicate this (lines 277-280 marked manuscript)

"The talus is predominantly composed of cobble to boulder-sized rocks, and geophysical surveys combined with hydrograph analysis indicate an openwork matrix with large void spaces without the presence of fine material (Muir et al., 2011). This coarse, blocky surface supports only sparse vegetation, making it difficult to infer thermally anomalous zones from vegetation patterns.."

*Equation makes the important assumption that the circulating air is assumed to be incompressible and the temperature is only a consequence of the density (if I understood it correctly). However, if the air in the soil/talus layers sinks downwards parallel to the slope (as indicated in Figures 5, 7 and 9), then it reaches regions of higher air pressure and undergoes compression, which is cannot be neglected over larger vertical distances (or conversely expansion as the air rises). Figure 5 also shows that the air in the talus body is in equilibrium with the outside air (i.e. corresponds to the respective air pressure). Was the assumption made here that the vertical movement is not large enough (maybe 10-20m) and therefore the vertical pressure change is not large enough and is neglected? However, this would limit the applicability of the model.*

The model employs the Oberbeck-Boussinesq approximation, which assumes constant air density in the air mass balance equation, while allowing for density variations in Darcy's law (Barletta, 2009; Landman and Schotting, 2007). This assumption is commonly applied in systems where pressure or temperature changes are not significant, simplifying the numerical solution of the problem.

Regarding external pressure, the model assumes that while external pressure does increase with elevation, it does not affect air density. For the relationship between external air pressure and internal air pressure, the model calculates the internal air pressure at each time step based on air

density (derived solely from temperature) and external air pressure (Eq. 6), which depends on elevation and acts as a boundary condition.

We acknowledge that neglecting air compressibility effects over large vertical gradients could limit the model's applicability. However, we believe this assumption is appropriate for the typical scenarios considered in our study. This simplification ensures that the model remains computationally efficient while providing reasonable results for the scale of the problem. We included the following text to indicate this (lines 164-175 marked manuscript):

"Analogous to the water mass balance, the air mass balance is formulated using Richards' equation, with the key distinction that air density is now temperature-dependent to account for buoyancy forces driving natural convection. The airflow equation is solved using the Oberbeck-Boussinesq approximation, which treats air flux as incompressible (Eq. 5) while retaining density variation solely within Darcy's law (Barletta, 2009; Landman and Schotting, 2007). This assumption is commonly applied in systems where pressure or temperature changes are not significant, simplifying the numerical solution. However, neglecting air compressibility may limit the model's applicability in cases dominated by strong vertical gradients. Regarding external pressure, the model assumes it increases with elevation and influences boundary conditions, but does not affect air density. As detailed in Appendix A2 the airflow equations are expressed as:"

*On page 18 the performance of GEOtop in comparison to the UEB model is described. It is stated that the GEOtop is well performing and show similar snow cover patterns as UEB. However, the difference of simulated SWE is rather large and also the time of maximum SWE is quite different. It seems that GEOtop is calibrated towards UEB in a way that it represents snow depletion well. I don't believe that differences in snow simulation change results in Figure 12 significantly. But I would rephrase the description of the performance of snow simulations of GEOtop more accurately according to the simulations.*

We rephrase the description of the performance of snow simulations (lines 440-444 marked manuscript):
"Figure 10B compares SWE from CE-GEOtop and UEB. Although there are differences in the magnitude and timing of peak SWE, both models capture similar snow depletion patterns, coinciding in the timing of snow-free ground. This suggests that CE-GEOtop can reasonably reproduce SWE dynamics in the Babylon talus slope, supporting its applicability in permafrost-related studies."

*Also the discharge simulations shown in Figure 12 are with significant deviations from observations (could add some skill measures here). This is explained in the text but it also stated that the GEOtop simulations align with observations in June and July. Though it is not easy to see exactly from Figure 12, it seems that also for June und July deviations are significantly (but the daily cycle is strong and give the impression of high coincidence).*

We computed the Nash–Sutcliffe model efficiency coefficient (NSE) based on daily values and obtained an overall NSE of 0.48 for the scenario without permafrost (CG-AO) and an overall NSE of 0.55 for the scenario involving permafrost (CG). According to Moriasi et. Al 2015, NSE values over 0.5 for daily streamflow values are considered satisfactory; therefore, following this value, the overall CG simulations give a satisfactory simulation of the observed daily streamflow during this period. If we calculate the NSE every 2 weeks. The NSE is only >0.5 for the period of the last week of June and the first week of July for the CG model.

To reflect the suggestions, we have rephrased the paragraph into (lines 507-512 marked manuscript):

"The Nash–Sutcliffe Efficiency (NSE) coefficient was calculated using daily discharge values, yielding an overall NSE of 0.48 for the CG-AO case and 0.55 for the CG case. According to Moriasi et al. (2015), daily NSE values greater than 0.5 are considered satisfactory, indicating that the model incorporating permafrost (CG case) provided a more accurate representation of basin discharge. Particularly, both cases aligned more closely with observed discharge during the last week of June and the first week of July. However, from mid-July onward, contributions from late-lying snowpack in zones not considered in the model became more significant (present in the upper part of the Babylon talus slope as depicted in Fig. 3) . As a result, the simulated discharge showed lower magnitudes than the observed values, matching the observed discharges only during precipitation events. Additionally, when comparing the spring temperatures to the measured data, it's evident that the temperatures from the model with permafrost closely resemble the measured values, while the case without permafrost overestimates the spring temperatures. These findings support the idea that the low water temperature during the summer may be related to the presence of coarse sediments and permafrost occurrence."

Moriasi, D. N., Gitau, M. W., Pai, N., & Daggupati, P. (2015). Hydrologic and water quality models: Performance measures and evaluation criteria. Transactions of the ASABE, 58(6), 1763-1785.

*Equations (13) and (14): wn is not used consistently here (does not appear in equation (13), but is stated afterwards)*

We revised equation 14 and included the following text to clarify this (line 233 marked manuscript):

" where $w_n$ are the weighting factors of air (a), ice (i), and solid particles, respectively.

*Figures 7 and 9: Please add a x-axis scaling, the arrows of air flow are hard to identify*

We have included a x-axis on both figures.

*The labelling of Figures 6, 8 and 11 could be improved to make it easier for the reader, e.g. indicate the profile name above the figure. A short description instead of the acronyms of the model configurations would also help (depending on the space available).*

We have moved the profile name to the top figures 6, 8, and 11. These figures were thought to be included using only one column in a two-column format, so a short description does not fit on the figure

*Figure 12: Please explain with more text*

The caption of Figure 12 (line 487 marked manuscript) was changed to:

" Effects of permafrost on discharge rates and spring water temperatures. (A) Hourly simulated and observed discharges. (B) Hourly simulated and observed temperatures. Pink lines show simulation results from the CE-GEOtop case with the air convection module turned off (CG-AO), while blue lines represent the CE-GEOtop case with air convection enabled (CG). In panel A, black lines indicate measured discharge; in panel B, black stars indicate measured water temperatures."

We thank the reviewers for their time in reviewing our paper. We provide responses to each individual point below. For clarity, comments are given in italics, and our responses are given in plain blue text.

*G. Zegers and colleagues enhanced GEOtop-3D, a well-known physically-based distributed hydrological model, to include the thermal effects of air convection and local thermal nonequilibrium (LTNE). Air convection exerts a cooling effect and critically shapes the permafrost distribution at landform scale, the scale needed for hydrological modelling in complex high-mountain catchments. The freely available model is (to our knowledge) the first geoscientific model that includes two-phase energy balance equations, four phases (rock-ice- water-air), and an explicit treatment of air and water flow. This is a valuable and important contribution and presented together with a very concisely written publication. The modelling is well connected to field observations and to the geomorphological literature where undercooling has been known for centuries. We have only few points to criticize.*

We appreciate the supportive comments.

*A major point is that the authors did not split in their paper results and discussion. This makes it difficult for the reader to see the results clearly separated from the interpretations. This is particularly true for Chapter 4.4 where the authors suddenly discuss the hydrological implications of permafrost on discharge. It would be nice to have the hydrological outcomes of this paper much better included in the whole paper, starting in the introduction and then also in the results and discussion section. We will give more details below (referring to L439).*

We addressed this concern in the detailed responses below and added a paragraph to the Introduction to better introduce the hydrological aspects. We also revised Section 4.4 to include additional detail and improve clarity in the interpretation of the results.

*Another point is that below-ground heat transfer by thermal radiation is ignored, which can be significant for cobble to boulder-sized rocks without fine sediments that would clog the large pore space. This has been shown in geotechnical experiments by Fillion et al. (2011) ("Radiation heat transfer becomes significant for d10 higher than 10 mm and predominant at values higher than 90 mm") and in the field by Scherler et al. (2014) and Amschwand et al. (2024). Radiation increases the stagnant thermal conductivity of the solid composite material. It does probably not change your main findings significantly, though, because the effective thermal conductivity values of ~1 W m71 K-1 would be again closer to those of the standard GEOtop implementation. Nonetheless, because large pores for radiation and high permeability for strong convection are likely to occur together in rocky debris, the issue should be briefly discussed (e.g ., in the Section at L191-218).*

We agree that incorporating thermal radiation could enhance the representation of the energy balance. However, it is beyond the scope of the present work. Accounting for thermal radiation in the current model setup is particularly challenging due to the presence of four distinct phases

(sediment, air, water, and ice), all of which need to be considered when estimating radiative heat transfer. We have mentioned thermal radiation at the end of the thermal conductivity subsection (lines 242-244 marked manuscript):

"One of the limitations of the CE-GEOtop model for simulating the thermal dynamics of coarse sediments is its exclusion of thermal radiation effects. Although some methods exist for estimating radiation in dry, coarse sediments (e.g., Fillion et al., 2011), the influence of water and ice content on thermal radiation requires further attention. "

*A few minor points are listed below.*

*Minor points L13: Please consider rephrasing to " ... local thermal equilibrium approaches underestimate the impact of natural convection especially on short timescales."*

We have rephrased (line 13 marked manuscript): to " ... local thermal equilibrium approaches underestimate the impact of natural convection, especially on short timescales."

*L28: Please change the reference from Riseborough et al. 2008 to a more specific reference like Van Everdingen, R. O ., 1998.*

We have changed the reference to van Everdingen, R. O ., 1998.

*L28/29: ... 'positive mean annual air temperature' this information was already explored far before the year 2000 as it is shown in your references. Already in the 1930s local researchers in Switzerland and elsewhere (e.g. Bächler 1930s) found such effects at certain places and in the early 1990s this was already scientifically described (e.g. Hoelzle 1992).*

We have included those references in the revised manuscript (line 29-30 marked manuscript).

*L40: You may add: https://doi.org/10.1002/esp.5998 (Wiegand & Kneisel ., 2024)*

We have included the reference in the revised manuscript (line 51 marked manuscript).

*L59: Suitable reference to add (and maybe to discuss) is Wicky et al. (2024), who also modelled air convection in a talus slope with a similar approach. Furthermore, please mention the works by Marchenko (2001) (a two-phase LTNE, 1-dim model where Rayleigh convection is parameterized), and Tanaka et al. (2000).*

We have included those references in the revised manuscript (line 29-30 marked manuscript).

*L112: Please consider rephrasing: " ... reducing the amount of energy gained by the ground from the atmosphere (Gruber and Hoelzle, 2008)"*

We have rephrased the line to: " ... reducing the amount of energy gained by the ground from the atmosphere (Gruber and Hoelzle, 2008)" (line 71 marked manuscript).

*L127: "Fw as a sink term representing evapotranspiration": Please clarify by saying that Fw affects the surface cells only (correct?) and is derived from the surface energy balance.*

Fw can act in the first layers of the sediments. Particularly, transpiration can go into deeper layers of the sediments depending on the model configuration. We have clarified this in the text (line 140 marked manuscript):

"Fw as a sink term representing evapotranspiration, which can act in the first layers of the sediments".

*L165: You could consider Amschwand et al. (2024, Fig. 4) for the closure of the snowpack.*

We have included Amschwand et al. (2024) in the mentioned references (line 184 marked manuscript).

*L170: Please consider rephrasing (see major point): "The energy transport equation in the air phase combines advection and conduction processes, but it does not account for the thermal radiation of coarse sediments". We suggest not to mention thermal radiation in the paragraph on the air energy equation (because air is essentially IR-transparent at such short distances), and to discuss it instead in the paragraph on the composite-medium conductivity (Sect. 2.3.2).*

We agree with the suggestion. We have mentioned thermal radiation in the thermal conductivity subsection (lines 242-244 marked manuscript):

"One of the limitations of the CE-GEOtop model for simulating the thermal dynamics of coarse sediments is its exclusion of thermal radiation effects. Although some methods exist for estimating radiation in dry, coarse sediments (e.g., Fillion et al., 2011), the influence of water and ice content on thermal radiation requires further attention. "

*L278: In the description of the Babylon talus slope, please mention the typical block size.*

We have included the following text (line 278 marked manuscript):

"The talus is predominantly composed of cobble to boulder-sized rocks, and geophysical surveys combined with hydrograph analysis indicate an openwork matrix with large void spaces without the presence of fine material (Muir et al., 2011).'

*L377: " ... air temperature, incoming longwave radiation, and cloud fraction (estimated from incoming shortwave radiation data (Long et al ., 2006))". It is not clear here how your input data is organized. Which radiation components have been measured, short-wave as well? Why did you use cloud fraction instead of the shortwave incoming radiation directly?*

Because incoming solar radiation is not constant in the domain and it depends on the shadow effect (Section 2.3.4), measured income solar radiation cannot be used directly. We have included the following text to clarify this (lines 414-418 marked manuscript):

"Due to the high spatial variability of solar radiation resulting from topographic shading (see Section 2.3.4), direct use of measured shortwave radiation is not feasible. Instead, half-hourly

cloud fraction is derived from measured incoming solar radiation (Long et al., 2006) and combined with hourly sun position and topographic shading to calculate pixel-specific incoming solar radiation across the model domain."

*L439: While Chapter 4.3 follows naturally from the previous discussion (permafrost distribution), Chapter 4.4 about the effects on discharge appears a bit sudden. A statement at the beginning of Chapter 4.4 or in the introduction about the links between ground thermal regime, ice content, and hydraulic properties (permeability) would make the transition clearer.*

We have included a paragraph in the introduction referring to permafrost and subsurface water flow (lines 33-39 marked manuscript):

"The presence of permafrost directly influences subsurface water flow. As soil reaches freezing temperatures, water becomes ice, hindering or blocking flow depending on the ice content. Permafrost generally acts as a confining layer, creating two main flow paths: near-surface flow above the permafrost (supra-permafrost) and deeper subsurface flow (sub-permafrost) (Giardino et al., 1992; Harrington et al., 2018; Langston et al., 2011; Woo et al., 1994. However, when permafrost is discontinuous or contains low ice content, intra-permafrost flow may also occur. Regarding the contribution of ice melt to streamflow, several studies have shown it is not highly significant in alpine landforms (Arenson and Jakob, 2010; Harrington et al., 2018; Hayashi, 2020; Krainer and Mostler, 2002). Thus, the primary function of permafrost in alpine hydrogeology is to control the subsurface water flow paths."

*While the interpretation of water temperatures (Fig. 12B) follows straightforward from the previous discussions, the discharge (Fig. 12A) is hard to interpret solely with the concepts presented in this paper. For example, the discharge is flashier in the CG-AO (no permafrost) model run than in the CG (permafrost) model run ("in the case with permafrost, discharge minimums tended to be higher, and the discharge maximums lower ,... "), which was attributed to refreezing and lower hydraulic permeability (L454). First: is this a conjecture or a model output?*

*Do these findings refer to deeper intra-talus water pathways? Please clarify, perhaps explaining (or showing) the different water pathways in the frozen/non-frozen talus.*

*Second, often the opposite is true, supra-permafrost runoff being more rapid and flashy than runoff in a non-permafrost catchment.*

This conclusion is based on an interpretation of the model results. The findings specifically refer to intra-talus water pathways. In the analyzed case, there is no supra-permafrost layer because the simulated permafrost is discontinuous with low ice content rather than a continuous body. As a result, there is no supra-permafrost flow. Instead, groundwater within the talus flows from more conductive layers toward less conductive but more saturated zones near the bedrock.

During June for the CG case, the snowmelt and infiltration led to an increase in ice content within the lower conductivity layer at the middle section of the talus (P3 profile). This increase is illustrated in the following figure (provided in the Appendix). We have included the following text to clarify this (lines 492-496 marked manuscript):

"These differences are attributable to intra-talus groundwater flow processes. The simulated permafrost within the talus slope is discontinuous and characterized by relatively low ice content, which prevents the formation of a continuous impermeable permafrost layer and consequently precludes supra-permafrost runoff. Thus, the hydrological behavior is dominated primarily by vertical infiltration through more conductive upper talus layers to deeper, less conductive but more saturated zones near the bedrock."

[Figure]

*L597: The links for the Leontini reference do not work.*

We apologize for the error in the original reference. The correct reference is "Su, Y., & Davidson, J. H. (2015). Modeling approaches to natural convection in porous media. Springer." https://link.springer.com/book/10.1007/978-3-319-14237-1. We have fixed this reference in the revised manuscript.

*L699: Please briefly comment on: How strong is the influence of moisture on the air density, and would taking the virtual temperature (or even the virtual potential temperature) improve the modelling?*

Moisture in the air has a minimal effect on air density at temperatures below 20 °C; therefore, including it would not significantly impact the simulation results. We have included the following text (lines 792-794 marked manuscript):

"This assumption is valid as, in this case, temperature gradients dominate the air fluxes, rather than air fluxes induced by water movement. Since moisture in the air has a minimal effect on air density at temperatures below 20 °C; this formulation also neglects the effect of moisture on air density."

*Figure 2: The figure is helpful. Please clarify that the CM energy balance is 1D vertical, or maybe draw 3D cubes instead of 2D squares.*

Yes, the CM energy balance is 1D vertical. We have changed the figure to 3D cubes instead of 2D squares, to clarify this.

*Figure 3: It is not clear why you do not use your GEOtop model to do the shortwave incoming radiation calculation as you explain also in the text (L311).*

You are right that the radiation calculation can be done directly with the GEOtop model however, the PISR values in this figure are shown as a qualitative analysis and cover a larger extent than one the employed GEOtop model. Nonetheless, the radiation calculations done by using SAGA GIS do consider the shadow effect in incoming radiation.

*In addition, your BTS-class - 3°C<BTS <- 2°C is not shown in the figure, but maybe there are no measurements within this BTS- class, please explain?*

Yes, there are no measurements within the BTS-class - 3°C<BTS <- 2°C.

*References*

*Amschwand, D., Scherler, M., Hoelzle, M., Krummenacher, B., Haberkorn, A., Kienholz, C., and Gubler, H.: Surface heat fluxes at coarse blocky Murtèl rock glacier (Engadine, eastern Swiss Alps), The Cryosphere, 18, 2103-2139, https://doi.org/10.5194/tc-18-2103-2024, 2024.*

*Bächler, E., 1930, Der verwünschte oder verhexte Wald im Brüeltobel: Appenzellerkalender, 209.*

*Fillion, M.-H., Côté, J., and Konrad, J.-M.: Thermal radiation and conduction properties of materials ranging from sand to rock-fill, Canadian Geotechnical Journal, 48, 532-542, https://doi.org/10.1139/t10-093, 2011.*

*Hoelzle, M., 1992, Permafrost occurrence from BTS measurements and climatic parameters in the eastern Swiss Alps: Permafrost and Periglacial Processes, v. 3, no. 2, p. 143-147.*

*Marchenko, S.: A model of permafrost formation and occurrences in the intracontinental mountains, Norsk Geografisk Tidsskrift - Norwegian Journal of Geography, 55:4, 230-234, DOI: 10.1080/00291950152746577, 2001.*

Scherler, M., Schneider, S., Hoelzle, M., and Hauck, C.: A two-sided approach to estimate heat transfer processes within the active layer of the Murtel-Corvatsch rock glacier, Earth Surf. Dynam., 2, 141-154, https://doi.org/10.5194/esurf-2-141-2014, 2014.

Tanaka, HL, Nohara, D., and Yokoi, D.: Numerical simulation of wind hole circulation and summertime ice formation at Ice Valley in Korea and Nakayama in Fukushima, Japan, Journal of the Meteorological Society of Japan. Ser. Il 78.5, 611-630, 2000. https://www.jstage.jst.go.jp/article/jmsj1965/78/5/78_5_611/_pdf/-char/ja http://gpvjma.ccs.hpcc.jp/~tanaka/web/papers/tnb2006_pdf.pdf

Van Everdingen, R. O., 1998, Multi-Language Glossary of Permafrost and related Ground-Ice Terms, in Association, I. P., ed.: Calgary, University Printing Services of the University of Calgary.

Wicky, J., Hilbich, C., Delaloye, R., and Hauck, C. (2024), Modeling the Link Between Air Convection and the Occurrence of Short-Term Permafrost in a Low-Altitude Cold Talus Slope. Permafrost and Periglac Process, 35: 202-217. https://doi.org/10.1002/ppp.2224